# The Impact of SARS-CoV-2 Infection on Heart Rate Variability: A Systematic Review of Observational Studies with Control Groups

**DOI:** 10.3390/ijerph20020909

**Published:** 2023-01-04

**Authors:** Chan-Young Kwon

**Affiliations:** Department of Oriental Neuropsychiatry, College of Korean Medicine, Dongeui University, 52-57, Yangjeong-ro, Busanjin-gu, Busan 47227, Republic of Korea; beanalogue@deu.ac.kr; Tel.: +82-51-850-8808

**Keywords:** COVID-19, SARS-CoV-2, HRV, vmHRV, RMSSD, HF

## Abstract

Autonomic nervous system (ANS) dysfunction can arise after severe acute respiratory syndrome coronavirus 2 (SARS-CoV-2) infection and heart rate variability (HRV) tests can assess its integrity. This review investigated the relationship between the impact of SARS-CoV-2 infection on HRV parameters. Comprehensive searches were conducted in four electronic databases. Observational studies with a control group reporting the direct impact of SARS-CoV-2 infection on the HRV parameters in July 2022 were included. A total of 17 observational studies were included in this review. The square root of the mean squared differences of successive NN intervals (RMSSD) was the most frequently investigated. Some studies found that decreases in RMSSD and high frequency (HF) power were associated with SARS-CoV-2 infection or the poor prognosis of COVID-19. Also, decreases in RMSSD and increases in the normalized unit of HF power were related to death in critically ill COVID-19 patients. The findings showed that SARS-CoV-2 infection, and the severity and prognosis of COVID-19, are likely to be reflected in some HRV-related parameters. However, the considerable heterogeneity of the included studies was highlighted. The methodological quality of the included observational studies was not optimal. The findings suggest rigorous and accurate measurements of HRV parameters are highly needed on this topic.

## 1. Introduction

The coronavirus disease of 2019 (COVID-19) resulting from a severe acute respiratory syndrome coronavirus 2 (SARS-CoV-2) infection has caused a serious public health crisis worldwide and has had harmful impacts on human health and quality of life [1]. SARS-CoV-2 infections can often lead to serious health consequences, including mortality, and are associated with multiple organ failures, including that of the respiratory, cardiovascular, nervous, hepatobiliary, immune, and blood systems, and mental health conditions such as brain fog [2,3,4].

Recent studies have reported an association between SARS-CoV-2 infection and autonomic dysfunction. For example, a systematic review by Scala et al. analyzed 22 studies reporting features of ANS involvement during acute COVID-19 and concluded that the disease had ANS involvement even at an early stage and the involvement was associated with poor prognosis of patients with COVID-19 [5]. As a mechanism to explain the relationship between SARS-CoV-2 infection and autonomic nervous system (ANS) function, Mohammadian et al. (2022) suggested that SARS-CoV-2 viral particles may disrupt the function of the ANS center in the brainstem by disrupting the homeostasis of the brain renin-angiotensin system [6]. Other researchers suggested the interplay of ANS function and inflammation [7]. Recently, ANS dysfunction in patients with COVID-19 has been shown to lead to the various symptoms seen in acute COVID-19 as well as long COVID [8]. In addition to SARS-CoV-2, ANS dysfunction may be part of the explanation for some symptoms in the context of other infectious diseases, including orthostatic intolerance in acquired immune deficiency syndrome, dyspnea in rabies, and impaired sweating function in leprosy [9].

Heart rate variability (HRV) tests measure the degree of variation between heartbeats, illustrating the body’s adaptability to distress and the integrity of ANS function [10,11]. Although there is still insufficient clinical evidence for it to be considered a routine test in clinical settings, the HRV test is non-invasive, easy to implement, and has been reconfirmed through clinical studies to estimate ANS dysfunction or mortality due to cardiovascular diseases such as arrhythmias, myocardial infarction, and congestive heart failure [12,13]. Since SARS-CoV-2 infection negatively affects the ANS and cardiovascular system [2] and causes psychological distress in COVID-19 patients [14], it is possible that the infection may be related to some outcomes of the HRV test. In addition, since proinflammatory conditions are associated with the prognosis and severity of COVID-19 [15], the known multifaceted interactions of the ANS and the immune system [16] suggest that HRV may be one of the important indicators of COVID-19. Since non-contact medical services are becoming increasingly necessary [17], the clinical relevance of HRV as a simple, non-invasive test must be studied. The standards for the confirmation of SARS-CoV-2 infection include detection of its antibody or antigen [18], and the HRV test has the potential to play a supplementary role. For example, there is a possibility that it can be used to indicate the need for tests to confirm SARS-CoV-2 infection in patients with mild or asymptomatic COVID-19, or to predict the prognosis of COVID-19 patients. Given that HRV is considered an important biomarker in the context of remote measurement technologies [19], its popularity has public health relevance to pandemics such as COVID-19.

However, the impact of SARS-CoV-2 infection on human HRV-related outcomes has not been comprehensively analyzed. Therefore, this study investigated the direct impact of SARS-CoV-2 infection on the results of HRV tests by comprehensively and systematically reviewing existing observational studies. We hypothesized that patients infected with SARS-CoV-2 show significant differences in HRV-related outcomes compared to non-infected individuals. There was also a hypothesis in this review that HRV parameters would be different depending on the severity and prognosis of COVID-19.

## 2. Materials and Methods

This protocol was registered in the Open Science Framework (registration number ZNX3V). There were no amendments after the protocol was registered. This systematic review complied with the Preferred Reporting Items for Systematic Reviews and Meta-Analyses (PRISMA) statement 2020 [20] (Appendix A).

### 2.1. Eligibility Criteria

#### 2.1.1. Types of Studies

Only observational studies with a control group, including observational cohort, cross-sectional, and case-control studies, were allowed. Interventional and other observational studies, including case reports and case series, were excluded. No restrictions were imposed on the publication language and publication status.

#### 2.1.2. Types of Participants

Only patients infected with SARS-CoV-2 were included. This review was intended to focus on the direct impact of SARS-CoV-2 infection in COVID-19 patients, and thus long COVID (i.e., the continuation or development of new symptoms 3 months after the initial infection with SARS-CoV-2) patients were excluded. This is because long COVID does not necessarily require the current presence of SARS-CoV-2, and the condition is not dependent on the severity of acute SARS-CoV-2 infection [21]. Therefore, the association between long COVID and HRV would be worthy of being investigated as a separate topic. No restrictions were imposed on clinical condition, language, sex/gender, age, or race/ethnicity.

#### 2.1.3. Types of Exposures

SARS-CoV-2 infection. There were no restrictions on the test for confirming the infection. However, the accuracy of the test was reflected in the quality assessment of the included studies.

#### 2.1.4. Types of Controls

Non-infected individuals, healthy controls, or infected patients of different severity of COVID-19 were included in the control group. For longitudinal studies, the time of infection, including pre-infection, was included. As a relationship between SARS-CoV-2 infection and HRV-related parameters has not yet been well established, this review focused on the impact of SARS-CoV-2 infection on HRV parameters. And comparative studies of COVID-19 patients and individuals with other infectious diseases were outside the scope of this review. Specifically, comparisons between COVID-19 patients and non-infected individuals or healthy controls investigated the association between SARS-CoV-2 infection and HRV parameters. On the other hand, comparisons among COVID-19 of different severities investigated the clinical usefulness of HRV in the context of the assessment of patients with COVID-19.

#### 2.1.5. Types of Outcome Measures

HRV-related outcomes included the following time and frequency domains. The time domains included the mean standard deviation of the normal-to-normal (NN) interval (SDNN), SDNN index, standard deviation of the average NN interval (SDANN), square root of the mean squared differences of successive NN intervals (RMSSD), standard deviation of the differences between adjacent NN intervals (SDSD), number of pairs of adjacent NN intervals differing by more than 50 ms in the entire recording (NN50 count), proportion of NN50 divided by the total number of NN intervals (pNN50), and HRV triangular index. The frequency domains included the mean total power (TP), powers in ultra-low frequency range (ULF), very low frequency range (VLF), low frequency (LF), coefficient of component variance for LF (CCVLF), high frequency (HF), coefficient of component variance for HF (CCVHF), normalized unit of LF power (LFnorm), normalized unit of HF power (HFnorm), and LF/HF ratio [10].

### 2.2. Search Strategy

Four electronic bibliographic databases were comprehensively searched by one researcher (C.-Y.K.). These databases included MEDLINE [via PubMed], EMBASE [via Elsevier], PsycARTICLES [via ProQuest], and the Cumulative Index to Nursing and Allied Health Literature [via EBSCO]. Moreover, an additional manual search on Google Scholar was conducted to search for gray and potentially missing literature. The search was conducted on 29 July 2022, and all studies published up to the search date were reviewed. The bibliographic information retrieved from each database was downloaded as a RIS format file and imported into EndNote20 (Clarivate Analytics, Philadelphia, PA, USA). The search strategy and search results from each database are included in Appendix A.

### 2.3. Study Selection

Two independent researchers (C.-Y.K. and B.L.) screened the titles and abstracts of documents to identify potential studies which meet the inclusion criteria. After the initial screening, the two independent researchers assessed the full texts of the screened studies. Any disagreements between the researchers were resolved through their discussion. EndNote20 (Clarivate Analytics, Philadelphia, PA, USA) was used to manage citations from included articles.

### 2.4. Data Extraction

Among the included studies, two independent researchers (C.-Y.K. and B.L.) extracted variables for analysis. These variables included first author, publication year, publication type, country, study type, comparison type, characteristics of participants, assessment tools for HRV, measured HRV parameters and findings, relationships between HRV parameters and other clinical variables, and author conclusions. Among the characteristics of the participants, in particular, the cardiovascular conditions of participants, which can be considered as an important covariate in the association between HRV parameters and COVID-19, was extracted. The extracted data from the included studies were entered into a Microsoft Excel file (Microsoft, Redmond, WA, USA). The two independent researchers performed the data extraction processes, and any inconsistencies were resolved through their discussion.

### 2.5. Quality Assessment

Depending on the study type, corresponding methodological quality assessment tools were used, developed by the National Heart, Lung, and Blood Institute group [22]. The Quality Assessment Tool for Observational Cohort and Cross-Sectional Studies was used for observational cohorts and cross-sectional study types. The Study Quality Assessment Tools Quality Assessment of Case-Control Studies was used for case-control studies. Two independent researchers (C.-Y.K. and B.L.) performed the quality assessment, and any disagreements were resolved through their discussion.

### 2.6. Data Analysis

Considering the heterogeneity of the population, times from the COVID-19 outbreak, and potential comorbid diseases, quantitative analysis was not planned in the protocol of this systematic review. A quantitative analysis can only be performed if sufficient homogeneity between the studies and outcomes used is ensured. However, the clinical heterogeneity between the included studies was considerable. Therefore, the impact of SARS-CoV-2 infection on HRV was analyzed qualitatively. Among the HRV parameters, clinically relevant vagally mediated HRV (vmHRV) parameters were of interest, which include RMSS, HF power, and HFnorm.

### 2.7. Assessment of Heterogeneity

The heterogeneity of included studies was investigated both qualitatively and quantitatively. Qualitatively, the potential causes of heterogeneity of the included studies according to study design, clinical characteristics of participants, and HRV measurement method were analyzed. Moreover, the authors performed a meta-analysis for the purpose of confirming the justification of performing only qualitative analysis. Command ‘metan’ in Software Stata version 13.1 (StataCorp, College Station, TX, USA) was used to perform meta-analysis. A meta-analysis was conducted on HRV parameters reported in two or more studies, in which value of HRV parameters were presented in the form of mean and standard deviation, and the number of subjects in each included group was specified. For the meta-analysis, a random-effect model was used, and the results were presented as standardized mean difference (SMD) and 95% confidence intervals (CIs). Instead of being interpreted clinically, the results of meta-analysis were interpreted for the purpose of visualizing and quantifying the heterogeneity of the study results. I-square statistic was used to estimate heterogeneity statistically, and if the values were greater than 50% and 75%, they were considered to have substantial and considerable heterogeneity, respectively.

## 3. Results

### 3.1. Study Selection

In the initial search, 259 documents were obtained, excluding 117 duplications. In the initial screening, 216 documents were excluded by their title and abstract. The inter-rater agreement rate among the researchers in this initial screening was 74.90% (194/259). After the initial screening, the full texts of the remaining 43 documents were reviewed. Three opinion articles [23,24,25], four review articles [6,26,27,28], two case reports or case series [29,30], one long COVID study [31], five studies that did not report HRV [32,33,34,35,36], eleven studies without a control group [37,38,39,40,41,42,43,44,45,46,47], and one study that used data from the journal article which was the same as the data presented in the conference abstract [48] were excluded. The inter-rater agreement rate among the researchers in the full texts review was 97.67% (42/43). Therefore, 17 observational studies (total participants, N = 3628) [49,50,51,52,53,54,55,56,57,58,59,60,61,62,63,64,65], including a study from other sources [61], were included in this review (Figure 1).

### 3.2. Characteristics of Included Studies

Of the seventeen included studies, three [49,56,57] were conference abstracts, and fourteen [50,51,52,53,54,55,58,59,60,61,62,63,64,65] were journal articles. Nine studies (52.94%) were conducted in the United States [52,53,54,56,57,58,62,63,65], two (11.76%) were conducted in Turkey [49,61], and the remaining studies were conducted in China [60], India [55], Italy [51], Serbia [59], Spain [50] and Switzerland [64]. A total of four study types were analyzed, including four case-control studies [49,55,59,65], six retrospective analyses [54,56,57,58,61,63], three cross-sectional studies [51,52,60], and four prospective cohort studies [50,53,62,64]. Eleven studies [49,51,52,53,55,57,58,59,61,62,65] included SARS-CoV-2 negative individuals or healthy controls as the control group. Eight studies included COVID-19 patients with different clinical manifestations (e.g., symptomatic or asymptomatic) [49,55,57], severity (e.g., mild or severe) [59,60], and clinical outcomes (e.g., mortality) [50,56,63] as a control group. Two studies included a self-control group, where Junarta et al. [54] compared pre-COVID-19 and post-COVID-19-related hospitalization in patients with chronic atrial fibrillation (cAF), and Risch et al. [64] compared changes in HRV parameters according to each stage in COVID-19 patients, including baseline, incubation, presymptomatic, symptomatic, and recovery stages. There were five studies [49,58,60,61,63] reporting history of cardiovascular conditions, and among them, four [49,60,61,63] reported that there was no statistically significant difference in cardiovascular condition between groups. Another three studies [50,51,55] excluded participants with concomitant conditions such as uncontrolled hypertension, arrhythmia, coronary artery disease, and cerebrovascular disease. Nine studies [49,51,54,55,56,57,60,61,65] used electrocardiograms (ECGs), four [52,53,62,64] used commercially available wearable devices (such as a Fitbit), two [59,63] used continuous monitoring tools for biometric information, and one [50] used the HRV-derived analgesia nociception index. In the remaining study [58], it was only described that a wearable wireless sensor was used (Table 1). The major findings of included studies are described in Appendix A.

### 3.3. Methodological Quality Assessment

#### 3.3.1. Case-Control Studies (n = 4)

In the four case-control studies [49,55,59,65], all of the research questions or objectives were clearly described. Three of these studies [55,59,65] clearly described the study population criteria, whereas the other study [49] did not. Of the case-control studies [49,55,59,65], none of them justified their sample size. In addition, all the included studies [49,55,59,65] did not describe whether the case group and control group were recruited from the same or similar population, including the timeframe. However, the case and control groups were clearly defined and separated. Among the studies [49,55,59,65], whether the protocol was previously registered was not clear. Therefore, it could not be confirmed whether the predefined inclusion and exclusion criteria were applied in these studies. Three studies [49,59,65] did not describe the sampling method, and the other study [55] noted that a convenient sampling method was used, but it was unclear whether randomization was applied or not. Among the studies [49,55,59,65], there was no mention of the use of concurrent controls. These studies included participants infected with SARS-CoV-2, but only three studies [55,59,65] clearly described the confirmation method. Although the assessment was not reported as being blind in these studies [49,55,59,65], since the confirmation of SARS-CoV-2 infection may have come from medical records, it was concluded that the identification of the participants may have been possible. All these studies reported that matched controls were included, but one study [49] did not describe the matching conditions. Regarding the overall quality, three [57,59,65] were evaluated as fair and the other study [49] as poor (Table 2).

#### 3.3.2. Other Types of Studies (n = 13)

In the remaining 13 other types of studies [50,51,52,53,54,56,57,58,60,61,62,63,64], including prospective cohort studies, cross-sectional studies, and retrospective reviews, all of the research questions were clearly described. However, three studies [51,57,58] did not clearly specify the study population. Four studies [52,53,60,62] did not describe information on eligible persons, and three studies [50,51,64] had a participation rate of more than 50% of eligible persons. In six retrospective chart reviews [54,56,57,58,61,63], the participation rate could not be evaluated. Except for these six retrospective studies [54,56,57,58,61,63], it was not possible to confirm whether the protocol was registered in advance for the other studies [50,51,52,54,60,62,64]. The sample size was justified in only two of these studies [50,64]. Infection with SARS-CoV-2 was confirmed in these studies [50,51,52,53,54,56,57,58,60,61,62,63,64] yet could not be assessed in varying amounts or levels. Except for three studies [54,58,63], the remaining studies specified the infection confirming method. In these studies [50,51,52,53,54,56,57,58,60,61,62,63,64], SARS-CoV-2 infection was confirmed once, and the outcome measures were clearly defined. Since the confirmation of SARS-CoV-2 infection may have come from medical records, it was concluded that the identification of the assessor might have been possible. Of the four studies that had follow-ups [50,53,62,64], only one [50] had a follow-up rate of less than 20%. Regression analysis was performed in only two studies [58,64], and statistical analysis considering potential covariates was not performed in the remaining studies. Regarding the overall quality, three [50,58,64] were evaluated as fair and the remaining ten studies [51,52,53,54,56,57,60,61,62,63] as poor (Table 3).

**Table 1 ijerph-20-00909-t001:** Characteristics of included studies.

Author (Country)	Study Type	COMPARISON	Population (IQR)	Assessment Tool (Duration)	HRV Parameters (Unit)
Sari 2020 (Turkey) [49]	case-control study	G1: symptomatic COVID-19 patients (n = 25) G2: asymptomatic COVID-19 patients (n = 25) G3: matched controls (n = 51)	G1 & G2: 38.5 ± 8.46 G3: 39.9 ± 15.3	ECG (Holter recording) (24 h)	1. SDNN (ms); 2. SDANN (ms); 3. RMSSD (ms); 4. SDNN Index; 5. pNN50 (%); 6. CCVLF; 7. CCVHF; 8. LF/HF
Aragón-Benedí 2021 (Spain) [50]	prospective cohort study	G1: critically ill COVID-19 who survived (n = 7) G2: critically ill COVID-19 who died (n = 7)	G1: 64 (60, 73) G2: 71 (57, 72)	HRV-derived analgesia nociception index (4 min)	1. Power (ms), index for SDNN; 2. mean ANI, index for HFnorm
Bellavia 2021 (Italy) [51]	cross-sectional study	G1: COVID-19 patients (n = 20) G2: COVID-19 negative controls (n = 20)	G1: 56.05 ± 19.15 G2: 52.55 ± 13.71	ECG (bipolar leads) (10 min in supine position, and 3 min during active standing)	1. SDNN (ms); 2. SDANN (ms); 3. pNN50 (%); 4. RMSSD (ms); 5. LF (ms^2^); 6. HF (ms^2^)
Gadaleta 2021 (USA) [52]	cross-sectional study	G1: COVID-19 patients (n = 198) G2: COVID-19 negative controls (n = 1614)	NR	Commercially available wearable device (Fitbit) (NR)	Average daily Z-score of HRV value
Hirten 2021 (USA) [53]	prospective cohort study	G1: COVID-19 patients (n = 13) G2: COVID-19 negative controls (n = 284)	36.3 ± 9.8	Commercially available wearable device (Apple watch) (NR)	1. MESOR of SDNN (ms); 2. mean amplitude of the circadian pattern of SDNN (ms); 3. mean acrophase of the circadian pattern of SDNN (ms)
Junarta 2021 (USA) [54]	retrospective review	G1(after): chronic atrial fibrillation + COVID-19 hospitalization G2(before): chronic atrial fibrillation + pre-COVID-19 Total N = 38	78.60 ± 11.37	ECG (NR)	1. SDSD (ms); 2. RMSSD (ms); 3. pNN50 (%)
Kaliyaperumal 2021 (India) [55]	case-control study	Analysis 1 G1: COVID-19 patients (n = 63) G2: matched controls (n = 43) Analysis 2 G3: symptomatic COVID-19 patients (n = 33) G4: asymptomatic COVID-19 patients (n = 33)	G1: 48.39 ± 16.3 G2: 50.1 ± 10.5 G3: 57.59 ± 13.5 G4: 38.57 ± 13.1	ECG (bipolar leads) (5 min)	1. HF (log data); 2. LF (ms^2^); 3. HF/LF; 4. LF/HF; 5. pNN50 (%); 6. RMSSD (log data); 7. SDNN (log data)
Kamaleswaran 2021 (USA) [56]	retrospective review	G1: critically ill COVID-19 who survived G2: critically ill COVID-19 who died Total N = 85	NR	ECG (5 min)	1. LF/HF; 2. VLF (ms^2^); 3. RMSSD (ms); 4. pNN50 (%)
Khalpey 2021 (USA) [57]	retrospective review	G1: symptomatic COVID-19 patients G2: asymptomatic COVID-19 patients with silent hypoxia G3: asymptomatic COVID-19 negative patients with silent hypoxia G4: symptomatic COVID-19 negative patientsTotal N = 200	NR	ECG (bipolar leads) (10 s)	1. RMSSD (ms); 2. SDNN (ms); 3. HRV triangular index
Lonini 2021 (USA) [58]	retrospective review	G1: COVID-19 patients (n = 15) G2: healthy controls (n = 14)	NR	Wearable wireless sensor (up to 2 min)	HRV (s)
Milovanovic 2021 (Serbia) [59]	case-control study	G1: mild COVID-19 patients (n = 30) G2: severe COVID-19 patients (n = 45) G3: matched controls (n = 77)	G1: M 40.71 ± 16.57, F 46.05 ± 16.78 G2: M 51.27 ± 17.60, F 52.18 ± 19.64 G3: M 44.11 ± 17.83, F 45.27 ± 18.94	Task Force© Monitor (device for continuous noninvasive hemodynamic and autonomic assessment) (NR)	1. LFnorm (nu); 2. HFnorm (nu); 3. VLF (ms^2^); 4. LF (ms^2^); 5. HF (ms^2^); 6. LF/HF
Pan 2021 (China) [60]	cross-sectional study	G1: mild COVID-19 patients (n = 13) G2: severe COVID-19 patients (n = 21)	G1: 47.5 ± 14.2 G2: 61.5 ± 15.0	ECG (24 h)	1. SDNN (ms); 2. SDANN (ms); 3. RMSSD (ms); 4. pNN50 (%); 5. LF (ms^2^); 6. HF (ms^2^); 7. LF/HF
Topal 2021 (Turkey) [61]	retrospective review	G1: confirmed COVID-19 patients (n = 53) G2: suspected COVID-19 patients (n = 42) G3: healthy controls (n = 20)	G1: 51.0 ± 13.1 G2: 53.6 ± 18.6 G3: NR	ECG (Holter recording) (24 h)	1. SDNN (ms); 2. SDNN index (ms); 3. SDANN (ms); 4. RMSSD (ms); 5. NN50 count; 6. pNN50 (%); 7. HRV triangular index; 8. LF (ms^2^); 9. HF (ms^2^); 10. LF/HF ratio
Hirten 2022 (USA) [62]	prospective cohort study	G1: COVID-19 patients (n = 49) G2: COVID-19 negative controls (n = 358)	G1: 37.3 ± 10.55 G2: 37.9 ± 9.73	Commercially available wearable device (Apple watch) (NR)	1. SDNN (ms); 2. HRV COSINOR parameters (MESOR, Amplitude, and Acrophase)
Ranard 2022 (USA) [63]	retrospective review	G1: COVID-19 patients with sudden cardiac death (n = 12) G2: COVID-19 patients without sudden cardiac death (n = 18)	G1: median 66 G2: median 73.5	Philips Intellivue MX800 monitors (5 min)	RMSSD (ms)
Risch 2022 (Switzerland) [64]	prospective cohort study	COVID-19 patients G1: Baseline; G2: Incubation; G3: Presymptomatic; G4: Symptomatic; G5: Recovery Total N = 66	43.66 ± 5.64	Commercially available wearable device (Ava-bracelet) (NR)	1. SDNN (ms); 2. RMSSD (ms); 3. LF/HF
Skow 2022 (USA) [65]	case-control study	G1: Omicron COVID-19 patients (n = 23) G2: matched controls (n = 13)	G1: 23 ± 3 G2: 26 ± 4	ECG (5 min)	1. HF (ms^2^); 2. LF (ms^2^)

Abbreviations: ANI, Analgesia nociception index; CCVHF, coefficient of component variance for high frequency; CCVLF, coefficient of component variance for low frequency; COVID-19, coronavirus disease of 2019; ECG, electrocardiography; HF, high frequency; HFnorm, normalized unit of high frequency power; HRV, heart rate variability; IQR, interquartile range; LF, low frequency; LFnorm, normalized unit of low frequency power; MESOR, midline statistic of rhythm; NN, normal-to-normal; NR, not reported; pNN50, the proportion of NN50 divided by the total number of NN intervals; RMSSD, the square root of the mean squared differences of successive NN intervals; SDANN, standard deviation of the average NN interval; SDNN, mean standard deviation of the NN interval; VLF, very low frequency.

**Table 2 ijerph-20-00909-t002:** Methodological qualities of included case-control studies.

Author	Q1	Q2	Q3	Q4	Q5	Q6	Q7	Q8	Q9	**Q10**	**Q11**	**Q12**
Sari 2020 [49]	Yes	No	No	NR	CD	Yes	NR	CD	Yes	No	No	CD
Kaliyaperumal 2021 [55]	Yes	Yes	No	NR	CD	Yes	CD	CD	Yes	Yes	No	Yes
Milovanovic 2021 [59]	Yes	Yes	No	NR	CD	Yes	NR	CD	Yes	Yes	No	Yes
Skow 2022 [65]	Yes	Yes	No	NR	CD	Yes	NR	CD	Yes	Yes	No	Yes

Abbreviations: CD, cannot determined; NR, not reported. Note. *Q1. Was the research question or objective in this paper clearly stated and appropriate? Q2. Was the study population clearly specified and defined? Q3. Did the authors include a sample size justification? Q4. Were controls selected or recruited from the same or similar population that gave rise to the cases (including the same timeframe)? Q5. Were the definitions, inclusion and exclusion criteria, algorithms or processes used to identify or select cases and controls valid, reliable, and implemented consistently across all study participants? Q6. Were the cases clearly defined and differentiated from controls? Q7. If less than 100 percent of eligible cases and/or controls were selected for the study, were the cases and/or controls randomly selected from those eligible? Q8. Was there use of concurrent controls? Q9. Were the investigators able to confirm that the exposure/risk occurred prior to the development of the condition or event that defined a participant as a case? Q10. Were the measures of exposure/risk clearly defined, valid, reliable, and implemented consistently (including the same time period) across all study participants? Q11. Were the assessors of exposure/risk blinded to the case or control status of participants? Q12. Were key potential confounding variables measured and adjusted statistically in the analyses? If matching was used, did the investigators account for matching during study analysis?* adopted from The National Heart, Lung, and Blood Institute [22].

**Table 3 ijerph-20-00909-t003:** Methodological qualities of included studies with other types.

Author	Q1	Q2	Q3	Q4	Q5	Q6	Q7	Q8	Q9	Q10	Q11	Q12	Q13	Q14
Aragón-Benedí 2021 [50]	Yes	Yes	Yes	CD	Yes	Yes	Yes	NA	Yes	No	Yes	No	Yes	No
Bellavia 2021 [51]	Yes	Yes	Yes	CD	No	Yes	Yes	NA	Yes	No	Yes	No	NA	No
Gadaleta 2021 [52]	Yes	No	CD	CD	No	Yes	Yes	NA	Yes	No	Yes	No	NA	No
Hirten 2021 [53]	Yes	Yes	CD	CD	No	Yes	Yes	NA	Yes	No	Yes	No	No	No
Junarta 2021 [54]	Yes	Yes	NA	NA	No	Yes	Yes	NA	No	No	Yes	No	NA	No
Kamaleswaran 2021 [56]	Yes	Yes	NA	NA	No	Yes	Yes	NA	Yes	No	Yes	No	NA	No
Khalpey 2021 [57]	Yes	No	NA	NA	No	Yes	Yes	NA	Yes	No	Yes	No	NA	No
Lonini 2021 [58]	Yes	No	NA	NA	No	Yes	Yes	NA	No	No	Yes	No	NA	Yes
Pan 2021 [60]	Yes	Yes	CD	CD	No	Yes	Yes	NA	Yes	No	Yes	No	NA	No
Topal 2021 [61]	Yes	Yes	NA	NA	No	Yes	Yes	NA	Yes	No	Yes	No	NA	No
Hirten 2022 [62]	Yes	Yes	CD	CD	No	Yes	Yes	NA	Yes	No	Yes	No	No	No
Ranard 2022 [63]	Yes	Yes	NA	NA	No	Yes	Yes	NA	No	No	Yes	No	NA	No
Risch 2022 [64]	Yes	Yes	Yes	CD	Yes	Yes	Yes	NA	Yes	No	Yes	No	No	Yes

Abbreviations: CD, cannot determined; NA, not applicable; NR, not reported. Note. *Q1. Was the research question or objective in this paper clearly stated? Q2. Was the study population clearly specified and defined? Q3. Was the participation rate of eligible persons at least 50%? Q4. Were all the subjects selected or recruited from the same or similar populations (including the same time period)? Were inclusion and exclusion criteria for being in the study prespecified and applied uniformly to all participants? Q5. Was a sample size justification, power description, or variance and effect estimates provided? Q6. For the analyses in this paper, were the exposure(s) of interest measured prior to the outcome(s) being measured? Q7. Was the timeframe sufficient so that one could reasonably expect to see an association between exposure and outcome if it existed? Q8. For exposures that can vary in amount or level, did the study examine different levels of the exposure as related to the outcome (e.g., categories of exposure, or exposure measured as continuous variable)? Q9. Were the exposure measures (independent variables) clearly defined, valid, reliable, and implemented consistently across all study participants? Q10. Was the exposure(s) assessed more than once over time? Q11. Were the outcome measures (dependent variables) clearly defined, valid, reliable, and implemented consistently across all study participants? Q12. Were the outcome assessors blinded to the exposure status of participants? Q13. Was loss to follow-up after baseline 20% or less? Q14. Were key potential confounding variables measured and adjusted statistically for their impact on the relationship between exposure(s) and outcome(s)?* adopted from The National Heart, Lung, and Blood Institute [22].

### 3.4. Impact on HRV Parameters

#### 3.4.1. HRV Parameters Investigated

There were a total of 17 HRV-related parameters investigated in the included studies, and the most commonly used index in 10 comparisons was the RMSSD (n = 17), followed by the SDNN (n = 15), LF/HF ratio (n = 14), LF power (n = 11), HF power (n = 11), and pNN50 (n = 9) (Appendix A).

#### 3.4.2. Impact on vmHRV Parameters

(1) SARS-CoV-2 infection (vs. negative control): Mixed results were observed for RMSSD. That is, two studies [55,61] found that the RMSSD (ms) in COVID-19 patients was statistically significantly lower than that of negative controls, while the other two studies [49,54] found the opposite significant result (all, *p* < 0.05). The other two studies [51,64] did not find a statistically significant difference on the parameter. In Risch et al. (2022) [64], no significant differences on RMSSD from baseline were found in the incubation, presymptomatic, symptomatic, and recovery stages of SARS-CoV-2 infection (all, *p* > 0.05). In Sari et al. (2020) [49], no difference in RMSSD between COVID-19 patients and negative controls was observed (*p* > 0.05), but the values of symptomatic COVID-19 patients were significantly lower than that of negative controls (*p* < 0.05).

Regarding SARS-CoV-2 infection, four studies [49,55,59,61] reported statistically significant reductions in HF power (ms^2^) compared to that of negative controls (all, *p* < 0.05). Although three studies [49,51,65] did not find a significant difference in this parameter between COVID-19 patients and negative controls, among the studies, Sari et al. (2020) [49] found that HF was significantly lower in symptomatic COVID-19 patients compared to negative controls (*p* < 0.05). Moreover, in Milovanovic et al. (2021) [59], mild COVID-19 patients showed lower HF values compared to negative controls (*p* < 0.05), but no significant difference was observed between severe COVID-19 patients and negative controls (*p* > 0.05). Regarding HFnorm (nu), Milovanovic et al. (2021) [59] found no statistically significant difference associated with SARS-CoV-2 infection (*p* > 0.05), but this parameter was rarely investigated in the included studies (Figure 2).

(2) Different severity of COVID-19: In relation to the different severity or prognosis of COVID-19, there were some studies that have found significant differences in vmHRV parameters between symptomatic and asymptomatic COVID-19 patients [49], and between died and survived COVID-19 patients [50,63]. Sari et al. (2020) [49] found that RMSSD and HF power were significantly lower in asymptomatic COVID-19 patients compared to symptomatic COVID-19 patients (both, *p* < 0.05). Aragón-Benedí et al. (2021) [50] found that the HFnorm was significantly higher in died COVID-19 patients compared to survived COVID-19 patients (*p* < 0.05). The remaining three studies [55,57,60] failed to find significant differences (Appendix A).

#### 3.4.3. Impact on Other HRV Parameters

(1) SARS-CoV-2 infection (vs. negative control): Regarding SDNN (ms), two studies [49,51] did not find significant changes in this parameter related to SARS-CoV-2 infection (both, *p* > 0.05). However, two other studies [55,61] found that this parameter was significantly lower in COVID-19 patients compared to negative controls (both, *p* < 0.05). Risch et al. (2022) [64] found that SDNN was significantly lower compared to baseline in all stages of incubation, presymptomatic, and symptomatic of COVID-19 (all, *p* < 0.05), except for the recovery stage (*p* > 0.05).

Mixed results were observed with respect to LF/HF ratio and LF power (ms^2^). Specifically, two included studies [59,61] found significantly higher LF/HF ratio in COVID-19 patients compared to negative controls, while another two [49,64] found a significantly lower ratio in COVID-19 patients (all, *p* < 0.05). For LF power, Topal et al. (2021) [61] found significantly higher LF in COVID-19 patients compared to negative controls, but two other studies [55,59] found the opposite significant result (all, *p* < 0.05).

Regarding pNN50 (%), three studies [49,51,55] did not find a significant difference between negative controls and COVID-19 patients (all, *p* > 0.05). However, among those studies, Sari et al. (2020) [49], comparing symptomatic COVID-19 patients and negative controls, found that the pNN50 was significantly lower in the former (*p* < 0.05). Junarta et al. (2021) [54] also found a significantly lower pNN50 in COVID-19 patients compared to negative controls (*p* < 0.05) (Appendix A).

(2) Different severity of COVID-19: Significant differences in some HRV parameters were found in comparisons between symptomatic and asymptomatic COVID-19 patients [49] and between severe and mild COVID-19 patients [60]. Specifically, Sari et al. (2020) [49] found that SDNN, pNN50, LF/HF ratio, and LF were significantly lower in symptomatic COVID-19 patients compared to asymptomatic COVID-19 patients (all, *p* < 0.05). Pan et al. (2021) [60] found significantly lower SDNN and significantly higher LF/HF ratio in severe COVID-19 patients compared to mild COVID-19 patients (both, *p* < 0.05). Aragón-Benedí et al. (2021) [50] comparing died and survived COVID-19 patients found no difference on SDNN between the groups (*p* > 0.05) (Appendix A).

### 3.5. Clinical Relevance of HRV Parameters in COVID-19 Patients

#### 3.5.1. ANS Imbalance of COVID-19 Patients

Common findings among the studies included a decrease in HRV or ANS activity [58,59,60,62,64] and an imbalance in the activity of the sympathetic and parasympathetic nervous systems [50,55,59,63] associated with SARS-CoV-2 infection or prognosis of COVID-19. Aragón-Benedí et al. (2021) [50] found that high HFnorm in COVID-19 patients indicated a depletion of sympathetic activity (*p* = 0.003), and Kaliyaperumal et al. (2021) [55] also found that parasympathetic activity was increased in COVID-19 patients based on the RMSSD > 40 ms (*p* = 0.01) and SDNN > 60 ms (*p* = 0.035). Though Milovanovic et al. [59] reported that severe COVID-19 patients had a significantly higher LF/HF ratio (*p* < 0.05), given the complexity of clinical interpretation of the ratio [66], it is difficult to interpret it as reporting inconsistent results with the previous two studies [50,55].

#### 3.5.2. Prognosis of COVID-19 Patients

Aragón-Benedí et al. (2021) [50] concluded that the high HFnorm found in COVID-19 patients was associated with a worse prognosis, higher mortality, and higher interleukin 6 (IL-6) levels. Specifically, they [50] found that COVID-19 patients with lower SDNN had a worse prognosis, including fewer survival days (*p* = 0.046), and higher HFnorm was correlated with higher IL-6 levels (*p* = 0.020). Ranard et al. (2022) [63] found that hospitalized patients with COVID-19 who experienced sudden cardiac death had a lower RMSSD and, therefore, lower parasympathetic activity (*p* < 0.0001). Kamaleswaran et al. (2021) [56] also found a statistically significant association between the RMSSD and survival of COVID-19 patients (*p* < 0.05).

### 3.6. Assessment the Heterogeneity

#### 3.6.1. Qualitative Analysis

The heterogeneity of the included studies was analyzed qualitatively in terms of the study design, clinical characteristics of participants, and HRV measurement method. The study designs of the included studies varied to a total of four (i.e., case-control study, retrospective analysis, cross-sectional study, and prospective cohort study), and only four included studies [50,53,62,64] (4/17, 23.53%) were prospective cohort studies with relatively strict study designs. In addition, only five studies [55,58,59,64,65] (5/17, 29.41%) were identified that statistically adjusted key potential confounding variables in their analysis.

Regarding the clinical characteristics of participants, most included studies [49,51,52,53,54,55,58,61,62,64,65] (10/17, 58.82%) recruited and classified participants according to SARS-CoV-2 positivity. However, they did not categorize according to COVID-19 severity, so it is possible that COVID-19 patients of varying severity were included. Meanwhile, some studies have classified COVID-19 patients according to clinical severity [59,60], recruited only critically ill patients [50,56], or classified participants with specific clinical signs including silent hypoxia [57] and death [50,63]. In addition, one study was interested in the occurrence of COVID-19 in cAF patients only [54]. One study [65] investigated the impact of the Omicron variant of COVID-19, but other studies did not limit the variant.

The HRV measurement method also showed differences according to the included studies. In particular, nine studies [49,51,54,55,56,57,60,61,65] (9/17, 52.94%) used ECG to measure HRV parameters, and the measurement time also varied from 10 s to 24 h. Only three studies [49,60,61] (3/9, 30%) described obtaining HRV parameters with 24 h of Holter recording. Meanwhile, four studies [52,53,62,64] (4/17, 23.53%) reported using commercially available wearable devices, and one study [58] reported using a wearable wireless sensor, but it was unclear whether they were commercially available (Table 1).

#### 3.6.2. Quantitative Analysis

Meta-analysis in comparison of COVID-19 positive patients and negative controls was possible in five HRV parameters from five studies [49,55,59,61,65], including two vmHRV (i.e., RMSSD and HF power), and three other parameters (i.e., SDNN, LF power, and LF/HF ratio). Meanwhile, meta-analysis in the comparison of symptomatic COVID-19 patients and asymptomatic controls was possible in two parameters from two studies [49,55], including SDNN and LF/HF ratio. Substantial visual and statistical heterogeneity (I-square values, 60.0% to 96.5%) was found in the meta-analysis results. Among the results of the meta-analysis, only the vmHRV parameter showed a statistically significant difference, that is, RMSSD (ms) was significantly greater (SMD, 1.15; 95% CIs, 0.60 to 1.71; *p* = 0.030) and HF power (ms^2^) was significantly lower (SMD, −1.49; 95% CIs, −2.64 to −0.34; *p* = 0.000) in COVID-19 patients compared to negative controls (Appendix A).

## 4. Discussion

### 4.1. Findings of This Review

This review was conducted to systematically investigate the effect of SARS-CoV-2 infection on HRV-related parameters, and a total of 17 observational studies [49,50,51,52,53,54,55,56,57,58,59,60,61,62,63,64,65] were included in this review. The main findings of this review are as follows:(1)Methodological quality of included studies: The methodological quality of the included observational studies was not optimal. Among the included studies, only two [50,64] justified the sample size, and in all studies, the blinding of analysis was not guaranteed. In addition, in studies other than case-control studies, there were only two studies [58,64] that measured potential confounding variables and adjusted statistically for their impact on the outcome.(2)HRV parameters investigated: The most frequently investigated HRV parameter in relation to SARS-CoV-2 infection was RMSSD, followed by LF/HF ratio, LF power, HF power, and pNN50.(3)Impact on vmHRV parameters: Among the significant differences found, compared to negative controls, a consistent finding of the vmHRV parameter associated with SARS-CoV-2 infection was low HF power. Mixed results were observed for RMSSD, and studies on HFnorm were lacking. In relation to the different severity or prognosis of COVID-19, it was reported that the RMSSD and HF power were significantly lower in symptomatic COVID-19 patients compared to asymptomatic COVID-19 patients, and that the RMSSD was significantly lower and the HFnorm was significantly higher in died COVID-19 patients than in survived patients. However, consistent findings were rare.(4)Impact on other HRV parameters: Some included studies have found significantly lower SDNN and pNN50 in patients with COVID-19 compared to negative controls, but mixed results were observed for LF/HF ratio and LF power. Significantly lower SDNN was observed in symptomatic patients compared to asymptomatic COVID-19 patients and in severe patients compared to mild COVID-19 patients. In the comparison of died and survived COVID-19 patients, no significant difference in SDNN was found, but low SDNN was significantly associated with worse prognosis of COVID-19 patients, including fewer survival days.(5)Heterogeneity of included studies: Included studies were heterogeneous in terms of study design, clinical characteristics of participants, and HRV measurement method. Also, in the quantitative analysis, substantial heterogeneity was observed for RMSSD, HF power, SDNN, LF power, and LF/HF ratio.

### 4.2. Clinical Interpretation

Although some included studies did not find a significant association between HRV-related parameters and SARS-CoV-2 infection, these studies did not deny the effect of SARS-CoV-2 infection on ANS function. For example, one study [51] found no significant differences in the HRV parameters between non-critically ill COVID-19 patients and healthy volunteers. However, this study [51] found significant differences between the two groups in Sudoscan and automated pupillometry results. Specifically, the patient group had significantly higher pupillary dilatation velocities (*p* = 0.040), baseline pupil diameter (*p* = 0.039), and incidence of feet sudomotor dysfunction (*p* = 0.038). The investigators [51] concluded the presence of ANS dysfunction in the early stage of COVID-19. Their findings may suggest a difference in sensitivity between the HRV test and other tests to observe changes in the ANS associated with SARS-CoV-2 infection.

The current review focused on the vmHRV parameter in relation to ANS function, to explore the association of parasympathetic activity to SARS-CoV-2 infection or COVID-19. As results, a significant drop in HF power related to SARS-CoV-2 infection was consistently observed [49,55,59,61]. This may be understood in the context of vagal invasion by SARS-CoV-2 and the important role of the nerve system in neuroimmunometabolism [67]. A potential association with the significant reduction in HF power observed in patients with long COVID can also be assumed [68], as the development of postural orthostatic syndrome after COVID-19 involves mechanisms such as increased sympathetic activity and decreased parasympathetic activity due to SARS-CoV-2 infection [69]. But the scope of this review limits the extension of the findings to long COVID. According to our findings, HFnorm showed no significant difference according to the presence or absence of SRAS-CoV-2 infection [59]. However, HFnorm was significantly higher in died COVID-19 patients, and the increase of this parameter was associated with a worse prognosis, higher mortality, and higher IL-6 levels [50]. They discussed this association by considering a high HFnorm a depletion of sympathetic activity and proportionally greater vagal activity [50]. The association of increased cardiac vagal and decreased cardiac sympathetic activities with death in patients with critically ill conditions such as acute respiratory distress syndrome was previously reported by the same research team [50]. The underlying mechanism explaining the association between poor prognosis and increased vagal modulation in critically ill patients needs further investigation, and may involve the use of anti-cholinergic and sympathomimetic agents used to correct ANS dysfunction in critically ill patients. RMSSD was significantly lower in symptomatic COVID-19 patients and deceased COVID-19 patients compared to controls [49]. Also, the decrease of this parameter was related to sudden cardiac death in hospitalized COVID-19 patients [63]. RMSSD, a parameter reflecting the integrity of vagus nerve-mediated autonomic regulation of the heart, is known to be negatively related to sudden death such as epileptic sudden death [70]. A lowered parameter suggests a reduced cardioprotective effect of the vagus nerve, possibly increasing the risk of myocardial damage and fatal arrhythmias [71].

Among the HRV parameters other than vmHRV, the significant differences observed were decreased SDNN, LF, HF, pNN50, and SDANN associated with SARS-CoV-2 infection or the poor severity and prognosis of COVID-19. Decreased SDNN and SDANN are consistent predictors of cardiac death [72,73], and this review also found that SARS-CoV-2 infection or severity was associated with reduced HRV [49,60,64]. In patients with sepsis, the SDNN was also classified as an effective parameter for predicting mortality in a previous meta-analysis [74]. Although not included in this review because of the lack of a control group, Mol et al. [43] concluded that a higher SDNN predicted survival, especially in elderly patients, in their retrospective cohort study.

These findings on HRV parameters suggest a potential involvement of vagal tone and ANS function in SARS-CoV-2 infection and the clinical course of COVID-19 patients, including symptom onset and death. However, since other confounding factors such as shifts in respiratory rate and volume may affect these parameters [10], and there were no included studies that strictly controlled for respiratory variables, the involvement of ANS is still tentative.

### 4.3. Limitations

Given the profound and widespread impact of COVID-19 on human health and the importance of non-invasive, early indicators, this systematic review has a strength as a first comprehensive review of the impact of SARS-CoV-2 infection on HRV-related variables. However, the following limitations are recognized. First, the heterogeneity of the studies included in this review is a major limitation of this review. Our review qualitatively analyzed the heterogeneity of the included studies, and found considerable heterogeneity in the study design, clinical characteristics of participants, and HRV measurement method. Among the included studies, the limitation that information on the cardiovascular conditions of the participants could be confirmed in only nine studies [49,50,51,54,55,58,60,61,63] (9/17, 52.94%) may also contribute to the heterogeneity. As such, only 53% of included studies [49,51,54,55,56,57,60,61,65] used ECG to measure HRV parameters, and it cannot be considered to have equal reliability as the HRV parameters obtained for commercially available wearable devices such as Fitbits. In addition, it is still possible that the heterogeneity was due to the age or sex of the participants, the difference in their underlying diseases, the difference in the duration of COVID-19, the difference in the history of COVID-19 vaccination, and the change in HRV due to factors other than SARS-CoV-2 infection. As the meta-analysis also reaffirmed the substantial heterogeneity included (I-square values, 60.0% to 96.5%), the meta-analysis was not interpreted clinically. Second, due to limitations in the methodological quality of the included studies, the reliability of the findings obtained in this review is challenged. In particular, a risk of bias was observed in justification of sample size, the possibility of assessor blinding, and analysis accounting potential covariates. Third, most of the included studies [49,50,51,54,56,58,60,63,64,65] were small, with a total sample size of fewer than 100 cases. The small sample sizes may have exaggerated the effect of SARS-CoV-2 infection on HRV-related variables. Moreover, the small sample sizes may explain the failure to find statistically significant differences between groups in most outcomes found in this review. Fourth, this review focused on SARS-CoV-2 infection and excluded long COVID; however, the clinical course of COVID-19 needs further refinement. Although one included study [64] divided the COVID-19 stage into five stages (i.e., baseline to recovery) and analyzed the change in HRV parameters for each stage, there is a limitation in that the other included studies did not specify a different period of COVID-19. Fifth, despite the well-known relationship between inflammation and some HRV parameters [75], there were no studies analyzing the relationship between HRV parameters and persisting inflammatory signs in patients with COVID-19. Sixth, studies on the threshold for detecting clinically meaningful changes, such as minimal clinically important differences in HRV parameters, are lacking. Therefore, although some changes of HRV parameters found in this review were statistically significant, they could not be rigorously interpreted as clinically meaningful changes. Finally, given that the scope of interest of this review was HRV parameters, suggesting some aspect of ANS function only, caution is needed in interpreting our findings to extend to the association between SARS-CoV-2 infection and ANS function.

## 5. Conclusions

The findings of this review show that SARS-CoV-2 infection, the severity of COVID-19, and its prognosis are likely to be reflected in some HRV-related parameters. Among vmHRV parameters, decreases in RMSSD and HF were associated with SARS-CoV-2 infection or the poor severity and prognosis of COVID-19, and decreases in RMSSD and increases in HFnorm were observed in died critically ill COVID-19 patients. However, this review highlights the considerable heterogeneity of the included studies. The findings suggest that rigorous and accurate measurements of HRV parameters are highly needed on this topic.

## Figures and Tables

**Figure 1 ijerph-20-00909-f001:**
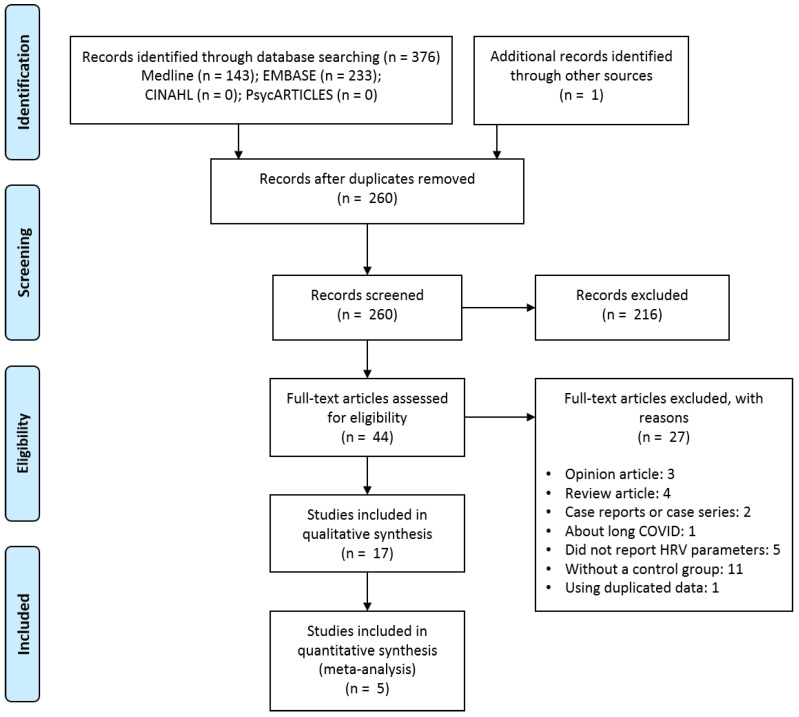
PRISMA flow diagram of this study. Abbreviations: CINAHL, Cumulative Index to Nursing and Allied Health Literature; COVID, coronavirus disease; HRV, heart rate variability.

**Figure 2 ijerph-20-00909-f002:**
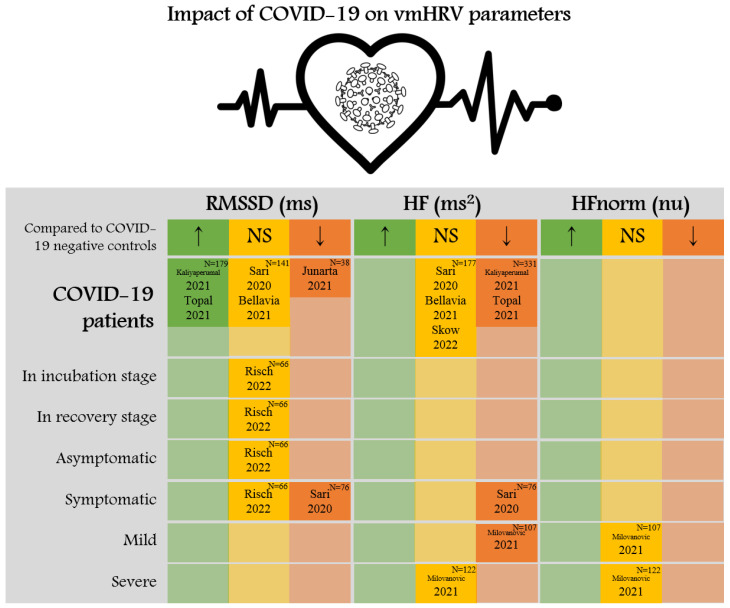
Impact of COVID-19 on vmHRV parameters [49,51,54,55,59,61,64,65]. Abbreviations: COVID-19, coronavirus disease of 2019; HF, high frequency; HFnorm, normalized unit of high frequency power; HRV, heart rate variability; NS, not significant; RMSSD, the square root of the mean squared differences of successive NN intervals; vmHRV, vagally mediated heart rate variability. Note: The number (N) in the upper right corner of each cell means the number of participants included in the analysis. ‘↑’ (green shading) and ‘↓’ (orange shading) mean significantly higher or lower than that of COVID-19 negative controls, respectively, while ‘NS’ (yellow shading) indicates no statistically significant difference was found.

## Data Availability

The data used to support the findings of this study are included within the article.

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
