# Peer review of "The Impact of SARS-CoV-2 Infection on Heart Rate Variability: A Systematic Review of Observational Studies with Control Groups"

_ijerph, 2023, doi:10.3390/ijerph20020909_

Round 1

Reviewer 1 Report

Review of manuscript  jerph-2059511

Dear authors,

I have read with interest your article entited: „The impact of SARS-CoV-2 infection on heart rate variability: a systematic review of observational studies with control groups“.In general it is a well written systematic review, the main clinical goal is however not easy to grasp. There are however some issues to be solved:

1)    Line 50-51, it holds true, that HRV can to some degree predict adverse cardiovascular events, but none of the test were meaningfull enough to enter clinical practice

2)    Line 57-58, I do not comprehend, to what extent a test like HRV may help the patients? The diagnosis is established by PCR test, you mean to risk stratify the patients? You have to explain what is the rationae behind the analysis! Otherwise it would be only descriptive statements. 

3)    Line 80, The sentence starting with: ,.. and thus long COVID patients…“ has to be rephrased. I do not understand what you mean by „long COVID“ , you have to explain!

4)    Line 86, it makes difference wether you include trials with non-infected/healthy individuals or different severity  SARSC-CoV-2 infected as controls. This has to be clarified, why you decided so. It may impose a significant flaw in the interpretetion of the results!

5)    SARS-CoV-2 infection increases also the risk of atrial arrhytmias, atrial fibrillation (Afib) beeing the most common one, are the included trials free of such a major flaw? Since Afib increases both the risk of cardiovascualr events and confounds any results from HVR analyis, because i tis irregular by nature. This shoulb be reflected both in the methods and discussion section!

6)    In the rsults section it has to be clear not only how many studies, but how many patients were included in the analyses!

7)    Line 355, in the discusion section i would suggest you summarize the main findings in the beginning of the section. i.e. The main findings of our systematic review analysis may be summarized as follows: 1) ..2)…. Etc.

8)    The conclusion sections has to be shortened and more to the point!I am not able to draw any conclusion from it by now.

Author Response

  • Response to Comments from Reviewer 1

General comment:

I have read with interest your article entited: „The impact of SARS-CoV-2 infection on heart rate variability: a systematic review of observational studies with control groups“.In general it is a well written systematic review, the main clinical goal is however not easy to grasp. There are however some issues to be solved:

Response:            

Thank you for your careful review and insightful comments that have significantly enhanced our manuscript. In this revision, we have tried to faithfully reflect the reviewer's comments.

Comment 1:

1)    Line 50-51, it holds true, that HRV can to some degree predict adverse cardiovascular events, but none of the test were meaningfull enough to enter clinical practice

Response:            

We agree with this comment. In this revised manuscript, we added an explanation that the relevant clinical evidence is still insufficient for HRV to be considered as a routine test in clinical practice.

“Heart rate variability (HRV) tests measure the degree of variation between heartbeats, illustrating the body's adaptability to distress and the integrity of ANS function [10,11]. Although there is still insufficient clinical evidence to be considered a routine test in clinical settings, the HRV test is non-invasive, easy to implement, and has been reconfirmed through clinical studies to estimate ANS dysfunction or mortality due to cardiovascular diseases such as arrhythmias, myocardial infarction, and congestive heart failure [12,13].”

(Please see in page 2, red words)

Comment 2:

2)    Line 57-58, I do not comprehend, to what extent a test like HRV may help the patients? The diagnosis is established by PCR test, you mean to risk stratify the patients? You have to explain what is the rationae behind the analysis! Otherwise it would be only descriptive statements.

Response:            

Thank you for the comment. As the reviewer commented, HRV test is unlikely to be conducive to the accuracy for diagnosis of COVID-19. However, it has the potential to play an auxiliary role around the PCR diagnosis, including predicting the infection of SARS-CoV-2 or predicting the prognosis of COVID-19. Moreover, since HRV parameter is one of the most used metrics for healthcare, it can be of great public health significance. Therefore, we have added the following sentences to this revised manuscript.

“Since non-contact medical services are becoming increasingly necessary [17], the clinical relevance of HRV as a simple, non-invasive test must be studied. The standards for the confirmation of SARS-CoV-2 infection include detection of its antibody or antigen [18], and the HRV test has the potential to play a supplementary role. For example, there is a possibility that it can be used to indicate the need for tests to confirm SARS-CoV-2 infection in patients with mild or asymptomatic COVID-19, or to predict the prognosis of COVID-19 patients. Given that HRV is considered an important biomarker in the context of remote measurement technologies [19], its popularity has public health relevance to pandemics such as COVID-19.”

(Please see in page 2, red words)

Comment 3:

3)    Line 80, The sentence starting with: ,.. and thus long COVID patients…“ has to be rephrased. I do not understand what you mean by „long COVID“ , you have to explain!

Response:            

Thank you for the comment. In this revised manuscript, we described the definition of long COVID and added why it is excluded from this review.

“2.1.2. Types of participants. Only patients infected with SARS-CoV-2 were included. This review was intended to focus on the direct impact of SARS-CoV-2 infection in COVID-19 patients, and thus long COVID (i.e., the continuation or development of new symptoms 3 months after the initial infection with SARS-CoV-2) patients were excluded. This is because long COVID does not necessarily require the current presence of SARS-CoV-2, and the condition is not dependent from the severity of acute SARS-CoV-2 infection [21]. Therefore, the association between long COVID and HRV would be worthy of being investigated as a separate topic. No restrictions were imposed on clinical condition, language, sex/gender, age, or race/ethnicity.”

(Please see in page 2, red words)

Comment 4:

4)    Line 86, it makes difference wether you include trials with non-infected/healthy individuals or different severity  SARSC-CoV-2 infected as controls. This has to be clarified, why you decided so. It may impose a significant flaw in the interpretetion of the results!

Response:            

Thank you for the comment. Like the reviewer's comment, the two topics (infection vs. non-infection; vs. different severity) can be considered separate. We reconstructed the manuscript by distinguishing these two topics.

“We hypothesized that patients infected with SARS-CoV-2 show significant differences in HRV-related outcomes compared to non-infected individuals. There was also a hypothesis in this review that HRV parameters would be different depending on the severity and prognosis of COVID-19.”

(Please see in page 2, red words)

“2.1.4. Types of controls. Non-infected individuals, healthy controls, or infected patients of different severity of COVID-19 were included in the control group. For longitudinal studies, the time of infection, including pre-infection, was included. As a relationship between SARS-CoV-2 infection and HRV-related parameters has not yet been well established, this review focused on the impact of SARS-CoV-2 infection on HRV parameters. And comparative studies of COVID-19 patients and individuals with other infectious diseases were outside the scope of this review. Specifically, comparisons between COVID-19 patients and non-infected individuals or healthy controls investigated the association between SARS-CoV-2 infection and HRV parameters. On the other hand, comparisons among COVID-19 of different severities investigated the clinical usefulness of HRV in the context of assessment of patients with COVID-19.”

(Please see in page 3, red words)

“3.4.2. Impact on vmHRV parameters

(1) SARS-CoV-2 infection (vs. negative control): Mixed results were observed …

(2) Different severity of COVID-19: In relation to the different severity or prognosis of COVID-19, there were some studies that have found significant …”

(Please see in page 11, red words)

“3.4.3. Impact on other HRV parameters

(1) SARS-CoV-2 infection (vs. negative control): Regarding SDNN (ms), …

(2) Different severity of COVID-19: Significant differences in some …”

(Please see in pages 12-13, red words)

Comment 5:

5)    SARS-CoV-2 infection increases also the risk of atrial arrhytmias, atrial fibrillation (Afib) beeing the most common one, are the included trials free of such a major flaw? Since Afib increases both the risk of cardiovascualr events and confounds any results from HVR analyis, because i tis irregular by nature. This shoulb be reflected both in the methods and discussion section!

Response:            

Thank you for the comment. We acknowledge that the original manuscript did not pay attention to the important covariate (i.e., cardiovascular conditions) in the association between HRV parameters and COVID-19. In this revised manuscript, we added the following in the Methods, Results, and Discussion sections regarding the cardiovascular conditions of the participants.

“2.4. Data extraction

Among the included studies, two independent researchers (CYK and BL) extracted variables for analysis. These variables included first author, publication year, publication type, country, study type, comparison type, characteristics of participants, assessment tools for HRV, measured HRV parameters and findings, relationships between HRV parameters and other clinical variables, and author conclusions. Among the characteristics of the participants, in particular, the cardiovascular conditions of participants, which can be considered as an important covariate in the association between HRV parameters and COVID-19, was extracted. The extracted data from the included studies were entered into a Microsoft Excel file (Microsoft, Redmond, WA, USA). The two independent researchers performed the data extraction processes, and any inconsistencies were resolved through their discussion.”

(Please see in pages 3-4, red words)

“Two studies included a self-control group, where Junarta et al. [54] compared pre-COVID-19 and post-COVID-19-related hospitalization in patients with chronic atrial fibrillation (cAF), and Risch et al. [64] compared changes in HRV parameters according to each stage in COVID-19 patients, including baseline, incubation, presymptomatic, symptomatic, and recovery stages. There were five studies [49,58,60,61,63] reporting history of cardiovascular conditions, and among them, four [49,60,61,63] reported that there was no statistically significant difference in cardiovascular condition between groups. Other three studies [50,51,55] excluded participants with concomitant conditions such as uncontrolled hypertension, arrhythmia, coronary artery disease, and cerebrovascular disease.”

(Please see in page 5, red words)

“However, the following limitations are recognized. First, the heterogeneity of the studies included in this review is a major limitation of this review. Our review qualitatively analyzed the heterogeneity of the included studies, and found considerable heterogeneity in the study design, clinical characteristics of participants, and HRV measurement method. Among the included studies, the limitation that information on the cardiovascular conditions of the participants could be confirmed in only nine studies [49-51,54,55,58,60,61,63] (9/17, 52.94%) may also contribute to the heterogeneity.”

(Please see in page 16, red words)

Comment 6:

6)    In the rsults section it has to be clear not only how many studies, but how many patients were included in the analyses!

Response:            

Thank you for the comment. In this revised manuscript, the number of total patients were described.

“Therefore, 17 observational studies (total participants, N = 3628) [49-65], including a study from other sources [61], were included in this review (Figure 1).”

(Please see in page 4, red words)

Comment 7:

7)    Line 355, in the discusion section i would suggest you summarize the main findings in the beginning of the section. i.e. The main findings of our systematic review analysis may be summarized as follows: 1) ..2)…. Etc.

Response:            

Thank you for the comment. Based on the comment, we divided the contents into (1), (2), and (3) and described them.

“(1) Methodological quality of included studies: The methodological …

(2) HRV parameters investigated: The most frequently …

(3) Impact on vmHRV parameters: Among the …

(4) Impact on other HRV parameters: Some included …

(5) Heterogeneity of included studies: Included studies …”

(Please see in page 14, red words)

Comment 8:

8)    The conclusion sections has to be shortened and more to the point!I am not able to draw any conclusion from it by now.

Response:            

Thank you for the comment. The conclusions have been further summarized and shortened.

“The findings of this review show that SARS-CoV-2 infection, the severity of COVID-19, and its prognosis are likely to be reflected in some HRV-related parameters. Among vmHRV parameters, decreases in RMSSD and HF were associated with SARS-CoV-2 infection or the poor severity and prognosis of COVID-19, and decrease in RMSSD and increase in HFnorm were observed in died critically ill COVID-19 patients. However, this review highlights the considerable heterogeneity of the included studies. The findings suggest rigorous and accurate measurements of HRV parameters are highly needed on this topic.”

(Please see in page 16, red words)

Reviewer 2 Report

The authors present a systematic review & meta analysis on the influence of SARS-CoV-2 infection on measures of HRV. They include observational studies with control groups. Overall the review is pre-registered and the protocol is largely followed, a PRISMA checklist is available. Well done so far, one of the best sys reviews regarding reporting of methods & transparency. 

I have some suggestions ordered in the line of appearance:

Line 32: add mental health, brain fog

Line 43: please stick to the definition of post covid by using WHO-definition

Line 51 Lately there has been a larger Meta-Analysis on on lower measures of HRV predicting allcause and cardiac Mortality in Neuroscience and Biobehavioural reviews

Line 68: OSF pre-registration & Prisma: well done

Line 79: Were there any limits to the type of diagnosis in terms of how the SARS-CoV-2  infection was approved? I.e. only positive PCR, or any rapid antigen test? Only physician diagnosed infections? How did you handle C19-vaccination status in the studies?? 

Line 106: I wondered why you did not use (“SARS-CoV-2"[Mesh] or "COVID-19"[Mesh]) instead or additional to the [MH]. Another HRV search term is “heart period variation”.  Maybe useful for future searches.

Line 112: Did you use any screening assistance system like https://www.rayyan.ai , research rabbit ? Please describe your workflow, i.e. where did the results from the the search engines were imported,  and handled. 

Line 115 & 122: Can you indicate an inter rater agreement / percentage how many papers needed to be discusesd?

Line 146: The nature of your research question is already pretty broad, a meta -analysis needs more specifically focus (I.e. measure, design). Maybe in a later stage and different publication authors can summarize their findings in a meta -analysis. This would be of interest to the public in my opinion. 

Line 225: Confirmation of COVID-19 (the disease) is different from confirmation of SARS-CoV-2 While reading 3.3.2 I was not sure if the authors mean SARS infection instead of COVID disease. 

Line 238: can you explain SDNN5 parameter in your manuscript 2.1.5. &  below the table1? Not all parameters included in Fig2 & tab1 are listed in section 2.1.5. please complete this section 

Line 247 & 260: If that are the tables from the NHLBI , please add a short note about the source or include a citation to the table footer (e.g.  adopted from NHLBI [17])

Line 283: Despite statistical significance, is it possible to infer if there were clinically meaningful differences in HRV change? 

Line 321: This is an important section. One factor not considered is breathing frequency, shaping not only distribution of parameters but also the ratios. Most studies will not have reported Breathing frequency I guess, bit you should definitely discuss the speciality of covid19 (breathing frequencies higher vs normal and its impact on HRV). It is also a source of between study heterogeneity, that you may add to your list in the discussion section. 

Line 330: LF/HF is not so much an indicator of sympathetic vs. Vagus activity. Rather, in the LF its a mixed of baroreceptor, breathing and temperature (see your Ginsberg ans Shaffer reference for more details for interpretation of LF/HF. 

Line 332:  please add the unit “ms” to RMSSD >40 & SDNN >60

Line 336: Can you provide details what makes the studies so different from each other? Could it be something like symptom severity , age etc?  Of course, a meta-regression is not possible, but from your impression is there something / do you have a guess?

In your registration you stated that u will contact authors for missing information. Did you contact any? I assume that there is a publication bias in terms of underreporting of insignificant HRV parameters. Please add the info how many were contacted, how many did answer and discuss this risk of bias source.

Line 345: The quality of Figure 2 is not sufficient in the copy that I have received. It looked like a screenshot that was enlarged. Please provide a higher quality image of the table. The color “Gray” is not included in the legend, please add. Please superscript  the 2 in ms2 in the according frequency domain parameters.  I am struggling with the color coding, at a first glance I thought orange and red belong together.  You are trying to transport a lot information in one table. Have u tried to place the author in the first column, the comparison in the second and then use arrows up and down to indicate the effects by HRV marker?  You may also colorcode the cells.  And you may also leave out markers that are reported in a single study only and move them them to the appendix (Just a suggestion to better focus the table).  Also right now you sorted the HRV markers by the most frequent ones to the left. I could imagine that a clustering of vagally mediated vs. Mixed measures may help to capture the specificity of covid on vagus better. 

This figure is so important for your conclusion(s). Did you contact study authors to retrieve other parameters?  Very likely they will report (no significant differences) but ask them  for the values to be able to run a meat analysis later (see comment above for Line 146)

Line 411: You describe the exposure differences as source of heterogeneity, but population characteristics and variation in measurement of the HRV outcome(s) likely added to heterogeneity, too. Please add. —> ok I see it is described later in line 436. You may add the dimensions of recording length and posture.

Line 538: reference 17: accessed on “” date is missing, please add

Please add to your supplemental files a header including the authors, title and journal that theses files belong to (Or DOI if available). And include page numbers. I suggest to upload the supplemental files as PDFs to the MDPI system. 

Could you add to the tables and supplement tables the reference number additional to author, year & location? It is hard to follow the in text citations in numbers and connect them to the tables using author year format. 

Forest plots in S4:

U can adopt the Stata code below  to better structure your code in the “Condition” variable and your 

Left hand column (locks option in Stata metan). First author is a string variable containing study name & year

Info is  a string contains a description (i.e. condition, sample size )

Usually, the central indicator includes a weight information (I.E. size of marker = relative size within studies)

#delimit ;

metan  lnhr lnsehr ,  npts(ngesamt)   random eform effect(“SMD”) 

by(HRV) nooverall sortby(sorter periodhrvanalyzed firstauthor)

forestplot(xlabel(-2.5  (0.5) 2.5, force)  

lcols(firstauthor   info )

boxopt(mcolor(gs13)) nowarning

diamopt(lcolor(navy))

pointopt( msymbol(T) mcolor(navy) msize(tiny))

ciopt( lcolor(gs8) lwidth(medium) )

olineopt(lcolor(navy) lpattern(dash))

favours("Favours lower HRV  "  #   "Favours higher HRV  "  )

title(“SDNN (ms)”, size(small))

graphregion(fcolor(white) lcolor(gs15) ifcolor(white) ilcolor(white))) ;

#delimit cr

Author Response

  • Response to Comments from Reviewer 2

General comment:

The authors present a systematic review & meta analysis on the influence of SARS-CoV-2 infection on measures of HRV. They include observational studies with control groups. Overall the review is pre-registered and the protocol is largely followed, a PRISMA checklist is available. Well done so far, one of the best sys reviews regarding reporting of methods & transparency.

I have some suggestions ordered in the line of appearance:

Response:            

Thank you for your careful review and insightful comments that have significantly enhanced our manuscript.

Comment 1:

Line 32: add mental health, brain fog

Response:            

Thank you for the comment. We added content on mental health.

“SARS-CoV-2 infections can often lead to serious health consequences, including mortality, and are associated with multiple organ failures, including that of the respiratory, cardiovascular, nervous, hepatobiliary, immune, and blood systems, and mental health condition such as brain fog [2-4].”

(Please see in page 1, red words)

Comment 2:

Line 43: please stick to the definition of post covid by using WHO-definition

Response:            

Thank you for the comment. Throughout this revised manuscript, we have reviewed and modified terminology for COVID-19-related conditions, such as long COVID, to align with the WHO definition.

“2.1.2. Types of participants. Only patients infected with SARS-CoV-2 were included. This review was intended to focus on the direct impact of SARS-CoV-2 infection in COVID-19 patients, and thus long COVID (i.e., the continuation or development of new symptoms 3 months after the initial infection with SARS-CoV-2) patients were excluded.”

(Please see in page 2, red words)

Comment 3:

Line 51 Lately there has been a larger Meta-Analysis on on lower measures of HRV predicting allcause and cardiac Mortality in Neuroscience and Biobehavioural reviews

Response:            

Thank you for the comment. In this revised manuscript, the study recommended by the reviewer is cited.

“Although there is still insufficient clinical evidence to be considered a routine test in clinical settings, the HRV test is non-invasive, easy to implement, and has been reconfirmed through clinical studies to estimate ANS dysfunction or mortality due to cardiovascular diseases such as arrhythmias, myocardial infarction, and congestive heart failure [12,13].”

[13] Jarczok, M.N.; Weimer, K.; Braun, C.; Williams, D.P.; Thayer, J.F.; Gündel, H.O.; Balint, E.M. Heart rate variability in the prediction of mortality: A systematic review and meta-analysis of healthy and patient populations. Neurosci Biobehav Rev 2022, 143, 104907, doi:10.1016/j.neubiorev.2022.104907.

(Please see in page 2, red words)

Comment 4:

Line 68: OSF pre-registration & Prisma: well done

Response:            

Thank you.

Comment 5:

Line 79: Were there any limits to the type of diagnosis in terms of how the SARS-CoV-2  infection was approved? I.e. only positive PCR, or any rapid antigen test? Only physician diagnosed infections? How did you handle C19-vaccination status in the studies??

Response:            

Thank you for the comment. This review did not place restrictions on the method of SARS-CoV-2 confirmation. However, the method of confirmation in the included studies was reflected in the assessment of their methodological quality.

“2.1.3. Types of exposures. SARS-CoV-2 infection. There were no restrictions on the test for confirming the infection. However, the accuracy of the test was reflected in the quality assessment of the included studies.”

(Please see in page 3, red words)

Comment 6:

Line 106: I wondered why you did not use (“SARS-CoV-2"[Mesh] or "COVID-19"[Mesh]) instead or additional to the [MH]. Another HRV search term is “heart period variation”.  Maybe useful for future searches.

Response:            

Thank you for the comment.

1) As far as we know, MH and MeSH are the same search tag. For example, a search with MeSH and MH for COVID-19 and SARS-CoV-2 yields the same result: (search date: 2022.12.22.)

2) Thank you for recommending the search term “heart period variation”. As of today, we searched for “heart period variation” on PubMed, and none of them were related to COVID-19 (i.e., only 7 results were found. Search date: 2022.12.22.), but if there are update searches in the future, I will include the term in our search strategy.

Comment 7:

Line 112: Did you use any screening assistance system like https://www.rayyan.ai , research rabbit ? Please describe your workflow, i.e. where did the results from the the search engines were imported,  and handled.

Response:            

Thank you for the comment. We used EndNote20 to conduct this systematic review and managed the bibliographic information with this program. Meanwhile, we took this opportunity to learn about the screening assistance system called ‘rayyan’ thanks to you. This is expected to facilitate future systematic reviews of our team.

“Four electronic bibliographic databases were comprehensively searched by one researcher (CYK). These databases included MEDLINE [via PubMed], EMBASE [via Elsevier], PsycARTICLES [via ProQuest], and the Cumulative Index to Nursing and Allied Health Literature [via EBSCO]. Moreover, an additional manual search on Google Scholar was conducted to search for gray and potentially missing literature. The search was conducted on July 29, 2022, and all studies published up to the search date were reviewed. The bibliographic information retrieved from each database was downloaded as a RIS format file and imported into EndNote20 (Clarivate Analytics, Philadelphia, USA). The search strategy and search results from each database are included in Supplementary file 2.

(Please see in page 3, red words)

Comment 8:

Line 115 & 122: Can you indicate an inter rater agreement / percentage how many papers needed to be discusesd?

Response:            

Thank you for the comment. We have added the rate.

“In the initial search, 259 documents were obtained, excluding 117 duplications. In the initial screening, 216 documents were excluded by their title and abstract. The inter-rater agreement rate among the researchers in this initial screening was 74.90% (194/259). After the initial screening, the full texts of the remaining 43 documents were reviewed. Three opinion articles [23-25], four review articles [6,26-28], two case reports or case series [29,30], one long COVID study [31], five studies that did not report HRV [32-36], 11 studies without a control group [37-47], and one study that used data from the journal article which was the same as the data presented in the conference abstract [48] were excluded. The inter-rater agreement rate among the researchers in the full texts review was 97.67% (42/43). Therefore, 17 observational studies (total participants, N = 3628) [49-65], including a study from other sources [61], were included in this review (Figure 1)”

(Please see in page 4, red words)

Comment 9:

Line 146: The nature of your research question is already pretty broad, a meta -analysis needs more specifically focus (I.e. measure, design). Maybe in a later stage and different publication authors can summarize their findings in a meta -analysis. This would be of interest to the public in my opinion.

Response:            

Thank you for the comment. Due to the lack of included studies and the heterogeneity of reported outcomes, the meta-analysis in this review was limited. Instead, we attempted to reflect this comment by adding a qualitative analysis of the heterogeneity.

“2.7. Assessment of heterogeneity

The heterogeneity of included studies was investigated both qualitatively and quantitatively. Qualitatively, the potential causes of heterogeneity of the included studies according to study design, clinical characteristics of participants, and HRV measurement method were analyzed. Moreover, the authors performed a meta-analysis for the purpose of confirming the justification of performing only qualitative analysis.”

(Please see in page 4, red words)

“3.6. Assessment the heterogeneity

3.6.1. Qualitative analysis. The heterogeneity of the included studies was analyzed qualitatively in terms of the study design, clinical characteristics of participants, and HRV measurement method. The study designs of the included studies varied to a total of four (i.e., case-control study, retrospective analysis, cross-sectional study, and prospective cohort study), and only four included studies [50,53,62,64] (4/17, 23.53%) were prospective cohort studies with relatively strict study designs. In addition, only five studies [55,58,59,64,65] (5/17, 29.41%) were identified that statistically adjusted key potential confounding variables in their analysis.

Regarding the clinical characteristics of participants, most included studies [49,51-55,58,61,62,64,65] (10/17, 58.82%) recruited and classified participants according to SARS-CoV-2 positivity. However, they did not categorize according to COVID-19 severity, so it is possible that COVID-19 patients of varying severity were included. Meanwhile, some studies have classified COVID-19 patients according to clinical severity [59,60], recruited only critically ill patients [50,56], or classified participants with specific clinical sign including silent hypoxia [57] and death [50,63]. In addition, one study was interested in occurrence of COVID-19 in cAF patients only [54]. One study [65] investigated the impact of the Omicron variant of COVID-19, but other studies did not limit the variant.

The HRV measurement method also showed differences according to the included studies. In particular, nine studies [49,51,54-57,60,61,65] (9/17, 52.94%) used ECG to measure HRV parameters, and the measurement time also varied from 10 seconds to 24 hours. Only three studies [49,60,61] (3/9, 30%) described obtaining HRV parameters with 24 h of holter recording. Meanwhile, four studies [52,53,62,64] (4/17, 23.53%) reported using commercially available wearable devices, and one study [58] reported using a wearable wireless sensor, but it was unclear whether they were commercially available (Table 1).

3.6.2. Quantitative analysis. Meta-analysis in comparison of COVID-19 positive patients and negative controls was possible in five HRV parameters from five studies [49,55,59,61,65], including two vmHRV (i.e., RMSSD and HF power), and three other parameters (i.e., SDNN, LF power, and LF/HF ratio). Meanwhile, meta-analysis in the comparison of symptomatic COVID-19 patients and asymptomatic controls was possible in two parameters from two studies [49,55], including SDNN and LF/HF ratio. Substantial visual and statistical heterogeneity (I-square values, 60.0% to 96.5%) was found in the meta-analysis results. Among the results of the meta-analysis, only the vmHRV parameter showed a statistically significant difference, that is, RMSSD (ms) was significantly greater (SMD, 1.15; 95% CIs, 0.60 to 1.71; p = 0.030) and HF power (ms2) was significantly lower (SMD, -1.49; 95% CIs, -2.64 to -0.34; p = 0.000) in COVID-19 patients compared to negative controls (Supplementary file 4).”

(Please see in pages 13-14, red words)

Comment 10:

Line 225: Confirmation of COVID-19 (the disease) is different from confirmation of SARS-CoV-2 While reading 3.3.2 I was not sure if the authors mean SARS infection instead of COVID disease.

Response:            

Thank you for the comment. As the reviewer points out, SARS-CoV-2 infection is a necessary but not sufficient condition for COVID-19. And the main topic of this review is not the relationship between COVID-19 and HRV, but the relationship between infection with SARS-CoV-2 and HRV. Nonetheless, this review also investigated the association between COVID-19 and HRV. This is because we also investigated differences in HRV parameters according to the severity of COVID-19. We have corrected relevant expressions in this revised manuscript to avoid misunderstanding.

“We hypothesized that patients infected with SARS-CoV-2 show significant differences in HRV-related outcomes compared to non-infected individuals. There was also a hypothesis in this review that HRV parameters would be different depending on the severity and prognosis of COVID-19.”

(Please see in page 2, red words)

“2.1.4. Types of controls. Non-infected individuals, healthy controls, or infected patients of different severity of COVID-19 were included in the control group. For longitudinal studies, the time of infection, including pre-infection, was included. As a relationship between SARS-CoV-2 infection and HRV-related parameters has not yet been well established, this review focused on the impact of SARS-CoV-2 infection on HRV parameters. And comparative studies of COVID-19 patients and individuals with other infectious diseases were outside the scope of this review. Specifically, comparisons between COVID-19 patients and non-infected individuals or healthy controls investigated the association between SARS-CoV-2 infection and HRV parameters. On the other hand, comparisons among COVID-19 of different severities investigated the clinical usefulness of HRV in the context of assessment of patients with COVID-19.”

(Please see in page 3, red words)

“3.4.2. Impact on vmHRV parameters

(1) SARS-CoV-2 infection (vs. negative control): Mixed results were observed …

(2) Different severity of COVID-19: In relation to the different severity or prognosis of COVID-19, there were some studies that have found significant …”

(Please see in page 11, red words)

“3.4.3. Impact on other HRV parameters

(1) SARS-CoV-2 infection (vs. negative control): Regarding SDNN (ms), …

(2) Different severity of COVID-19: Significant differences in some …”

(Please see in pages 12-13, red words)

Comment 11:

Line 238: can you explain SDNN5 parameter in your manuscript 2.1.5. &  below the table1? Not all parameters included in Fig2 & tab1 are listed in section 2.1.5. please complete this section

Response:            

Thank you for the comment. SDANN5 is actually the same as SDANN. This confusion has to do with the definition of SDANN, which is defined as ‘standard deviation of the average normal-to-normal (NN) intervals for each of the 5 minutes segments during a 24 h recording’. There were some studies that emphasized the 5 minutes and defined it as SDANN5. However, since this is the same as SDANN, in this revised manuscript, all SDANN5 are modified to SDANN. In this revised manuscript, all HRV parameters used in the included studies are defined in 2.1.5.

Comment 12:

Line 247 & 260: If that are the tables from the NHLBI , please add a short note about the source or include a citation to the table footer (e.g.  adopted from NHLBI [17])

Response:            

Thank you for the comment. We have added the citation to the foot note of these tables.

“…?, adopted from The National Heart, Lung, and Blood Institute [22].”

(Please see in pages 9-10, red words)

Comment 13:

Line 283: Despite statistical significance, is it possible to infer if there were clinically meaningful differences in HRV change?

Response:            

Thank you for the comment. As the reviewer points out, statistical significance does not imply clinical significance. In addition, minimal clinically important difference studies to prove clinical significance of HRV are also lacking. Therefore, we added the following as a limitation of this review.

Sixth, studies on the threshold for detecting clinically meaningful changes, such as minimal clinically important differences in HRV parameters, are lacking. Therefore, although some changes of HRV parameters found in this review were statistically significant, they could not be rigorously interpreted as clinically meaningful changes.”

(Please see in page 16, red words)

Comment 14:

Line 321: This is an important section. One factor not considered is breathing frequency, shaping not only distribution of parameters but also the ratios. Most studies will not have reported Breathing frequency I guess, bit you should definitely discuss the speciality of covid19 (breathing frequencies higher vs normal and its impact on HRV). It is also a source of between study heterogeneity, that you may add to your list in the discussion section.

Response:            

Thank you for the comment. This comment is specifically expert advice on this subject. HRV may be influenced by respiratory rate or volume, which should also be understood in the context of COVID-19. We added this to the Discussion section.

“These findings on HRV parameters suggest a potential involvement of vagal tone and ANS function in SARS-CoV-2 infection and the clinical course of COVID-19 patients, including symptom onset and death. However, since other confounding factors such as shifts in respiratory rate and volume may affect these parameters [10], and there were no included studies that strictly controlled for respiratory variables, the involvement of ANS is still tentative.”

(Please see in pages 15-16, red words)

Comment 15:

Line 330: LF/HF is not so much an indicator of sympathetic vs. Vagus activity. Rather, in the LF its a mixed of baroreceptor, breathing and temperature (see your Ginsberg ans Shaffer reference for more details for interpretation of LF/HF.

Response:            

Thank you for the comment. We agree that LF/HF ratio should not be interpreted simply as sympathetic versus vagal activity. In this revised manuscript, the interpretation of LF/HF has been reduced, and vmHRV has been highlighted instead.

“Considering the heterogeneity of the population, times from the COVID-19 outbreak, and potential comorbid diseases, quantitative analysis was not planned in the protocol of this systematic review. A quantitative analysis can only be performed if sufficient homogeneity between the studies and outcomes used is ensured. However, the clinical heterogeneity between the included studies was considerable. Therefore, the impact of SARS-CoV-2 infection on HRV was analyzed qualitatively. Among the HRV parameters, clinically relevant vagally mediated HRV (vmHRV) parameters were of interest, which include RMSS, HF power, and HFnorm.”

(Please see in page 4, red words)

“3.4.2. Impact on vmHRV parameters

(1) SARS-CoV-2 infection (vs. negative control): …”

(Please see in page 11, red words)

Figure 2. Impact of COVID-19 on vmHRV parameters.”

(Please see in page 12, red words)

“4.1. Findings of this review

This review was conducted to systematically investigate the effect of SARS-CoV-2 infection on HRV …

(3) Impact on vmHRV parameters: …”

(Please see in page 14, red words)

Comment 16:

Line 332:  please add the unit “ms” to RMSSD >40 & SDNN >60

Response:            

Thank you for the comment. We added it.

“Aragón-Benedí et al. (2021) [50] found high HFnorm in COVID-19 patients indicated a depletion of sympathetic activity (p = 0.003), and Kaliyaperumal et al. (2021) [55] also found that parasympathetic activity was increased in COVID-19 patients based on the RMSSD > 40 ms (p = 0.01) and SDNN > 60 ms (p = 0.035).”

(Please see in page 13, red words)

Comment 17:

Line 336: Can you provide details what makes the studies so different from each other? Could it be something like symptom severity , age etc?  Of course, a meta-regression is not possible, but from your impression is there something / do you have a guess?

Response:            

Thank you for the comment. We took this comment to more explore the heterogeneity qualitatively. Therefore, we further explored the heterogeneity between the studies, such as study design, participants, and HRV measurement methods. This was also reflected in our Methods, Results and Discussion sections.

“2.7. Assessment of heterogeneity

The heterogeneity of included studies was investigated both qualitatively and quantitatively. Qualitatively, the potential causes of heterogeneity of the included studies according to study design, clinical characteristics of participants, and HRV measurement method were analyzed. Moreover, the authors performed a meta-analysis for the purpose of confirming the justification of performing only qualitative analysis.”

(Please see in page 4, red words)

“3.6. Assessment the heterogeneity

3.6.1. Qualitative analysis. The heterogeneity of the included studies was analyzed qualitatively in terms of the study design, clinical characteristics of participants, and HRV measurement method. The study designs of the included studies varied to a total of four (i.e., case-control study, retrospective analysis, cross-sectional study, and prospective cohort study), and only four included studies [50,53,62,64] (4/17, 23.53%) were prospective cohort studies with relatively strict study designs. In addition, only five studies [55,58,59,64,65] (5/17, 29.41%) were identified that statistically adjusted key potential confounding variables in their analysis.

Regarding the clinical characteristics of participants, most included studies [49,51-55,58,61,62,64,65] (10/17, 58.82%) recruited and classified participants according to SARS-CoV-2 positivity. However, they did not categorize according to COVID-19 severity, so it is possible that COVID-19 patients of varying severity were included. Meanwhile, some studies have classified COVID-19 patients according to clinical severity [59,60], recruited only critically ill patients [50,56], or classified participants with specific clinical sign including silent hypoxia [57] and death [50,63]. In addition, one study was interested in occurrence of COVID-19 in cAF patients only [54]. One study [65] investigated the impact of the Omicron variant of COVID-19, but other studies did not limit the variant.

The HRV measurement method also showed differences according to the included studies. In particular, nine studies [49,51,54-57,60,61,65] (9/17, 52.94%) used ECG to measure HRV parameters, and the measurement time also varied from 10 seconds to 24 hours. Only three studies [49,60,61] (3/9, 30%) described obtaining HRV parameters with 24 h of holter recording. Meanwhile, four studies [52,53,62,64] (4/17, 23.53%) reported using commercially available wearable devices, and one study [58] reported using a wearable wireless sensor, but it was unclear whether they were commercially available (Table 1).

3.6.2. Quantitative analysis. Meta-analysis in comparison of COVID-19 positive patients and negative controls was possible in five HRV parameters from five studies [49,55,59,61,65], including two vmHRV (i.e., RMSSD and HF power), and three other parameters (i.e., SDNN, LF power, and LF/HF ratio). Meanwhile, meta-analysis in the comparison of symptomatic COVID-19 patients and asymptomatic controls was possible in two parameters from two studies [49,55], including SDNN and LF/HF ratio. Substantial visual and statistical heterogeneity (I-square values, 60.0% to 96.5%) was found in the meta-analysis results. Among the results of the meta-analysis, only the vmHRV parameter showed a statistically significant difference, that is, RMSSD (ms) was significantly greater (SMD, 1.15; 95% CIs, 0.60 to 1.71; p = 0.030) and HF power (ms2) was significantly lower (SMD, -1.49; 95% CIs, -2.64 to -0.34; p = 0.000) in COVID-19 patients compared to negative controls (Supplementary file 4).”

(Please see in pages 13-14, red words)

“(5) Heterogeneity of included studies: Included studies were heterogeneous in terms of study design, clinical characteristics of participants, and HRV measurement method. Also, in the quantitative analysis, substantial heterogeneity was observed for RMSSD, HF power, SDNN, LF power, and LF/HF ratio..”

(Please see in page 14, red words)

Comment 18:

In your registration you stated that u will contact authors for missing information. Did you contact any? I assume that there is a publication bias in terms of underreporting of insignificant HRV parameters. Please add the info how many were contacted, how many did answer and discuss this risk of bias source.

Response:            

Thank you for the comment. We sent e-mails to all authors except for three studies (Sari et al., Kamalewaran et al., and Khalpey et al.) for which contact e-mail addresses were not listed. The e-mail was asking for the presence of unreported HRV data in addition to the currently reported data.

We have yet to receive a reply from them, and we had to submit this revision today, the deadline extended at our request. But still, I will wait for a reply from them.

Comment 19:

Line 345: The quality of Figure 2 is not sufficient in the copy that I have received. It looked like a screenshot that was enlarged. Please provide a higher quality image of the table.

The color “Gray” is not included in the legend, please add.

Please superscript  the 2 in ms2 in the according frequency domain parameters.

 I am struggling with the color coding, at a first glance I thought orange and red belong together.

 You are trying to transport a lot information in one table. Have u tried to place the author in the first column, the comparison in the second and then use arrows up and down to indicate the effects by HRV marker?  You may also colorcode the cells.  And you may also leave out markers that are reported in a single study only and move them them to the appendix (Just a suggestion to better focus the table).

Also right now you sorted the HRV markers by the most frequent ones to the left. I could imagine that a clustering of vagally mediated vs. Mixed measures may help to capture the specificity of covid on vagus better.

This figure is so important for your conclusion(s). Did you contact study authors to retrieve other parameters?  Very likely they will report (no significant differences) but ask them  for the values to be able to run a meat analysis later (see comment above for Line 146)

Response:            

Thank you for the comment. We apologize for any inconvenience caused regarding the resolution of this image. We re-uploaded the image with improved resolution. Specifically, this revised figure focuses on vmHRV, and everything else has been moved to supplementary file 3. As we responded to the comment above, we asked the authors for their potentially unreported HRV parameters, but we did not receive a response.

Comment 20:

Line 411: You describe the exposure differences as source of heterogeneity, but population characteristics and variation in measurement of the HRV outcome(s) likely added to heterogeneity, too. Please add. —> ok I see it is described later in line 436. You may add the dimensions of recording length and posture.

Response:            

Thank you for the comment. the dimensions of recording length and posture It was also considered one of the causes of the heterogeneity found in the included studies.

“The HRV measurement method also showed differences according to the included studies. In particular, nine studies [49,51,54-57,60,61,65] (9/17, 52.94%) used ECG to measure HRV parameters, and the measurement time also varied from 10 seconds to 24 hours. Only three studies [49,60,61] (3/9, 30%) described obtaining HRV parameters with 24 h of holter recording.”

(Please see in pages 13-14, red words)

Comment 21:

Line 538: reference 17: accessed on “” date is missing, please add

Response:            

Thank you for the comment. The access date has been added.

Comment 22:

Please add to your supplemental files a header including the authors, title and journal that theses files belong to (Or DOI if available). And include page numbers. I suggest to upload the supplemental files as PDFs to the MDPI system.

Response:            

Thank you for the comment. I'm not sure if we understood this comment, but we created Supplementary 5. And the author, title, journal name, doi, and page number of the studies included in this review were described in this supplementary file.

(Please see Supplementary file 5)

Comment 23:

Could you add to the tables and supplement tables the reference number additional to author, year & location? It is hard to follow the in text citations in numbers and connect them to the tables using author year format.

Response:            

Thank you for the comment. We apologize for the inconvenience caused to the review due to insufficient reference citation. Reference numbers have been added to both tables and supplementary tables.

(Please see Supplementary files 3 and 5)

Comment 24:

Forest plots in S4:

U can adopt the Stata code below  to better structure your code in the “Condition” variable and your

Left hand column (locks option in Stata metan). First author is a string variable containing study name & year

Info is  a string contains a description (i.e. condition, sample size )

Usually, the central indicator includes a weight information (I.E. size of marker = relative size within studies)

#delimit ;

metan  lnhr lnsehr ,  npts(ngesamt)   random eform effect(“SMD”)

by(HRV) nooverall sortby(sorter periodhrvanalyzed firstauthor)

forestplot(xlabel(-2.5  (0.5) 2.5, force) 

lcols(firstauthor   info )

boxopt(mcolor(gs13)) nowarning

diamopt(lcolor(navy))

pointopt( msymbol(T) mcolor(navy) msize(tiny))

ciopt( lcolor(gs8) lwidth(medium) )

olineopt(lcolor(navy) lpattern(dash))

favours("Favours lower HRV  "  #   "Favours higher HRV  "  )

title(“SDNN (ms)”, size(small))

graphregion(fcolor(white) lcolor(gs15) ifcolor(white) ilcolor(white))) ;

#delimit cr

Response:            

Thank you for the comment. We specially thank you for this comment. Given the instructions provided by the reviewer, we performed a new meta-analysis and improved its readability.

(Please see in Supplementary file 4)

Reviewer 3 Report

1. Does the meta-analysis include the patient infection stage (short, long, acute...)? Why the long Covid-19 patents were excluded?

2. Which methods were used among studies for data extraction and filtration in HRV parameters (especially in FitBit and other non-medical devices)? Are the data comparable in terms of quality?

3. The Results section is not systematized. The reader encounters many opposite results, comparing study by study. Results, Discussion, and Conclusion must be systematized in order to address some of the main criteria of SARS-CoV-2 infection impact on HRV among selected studies.

The main results and differences among time and frequency HRV domains should be highlighted.

What is the common denominator of all the studies?

Heterogeneity did not bring clarity to the presentation of the results. 

4. The methodological research does not include studies like the following:

https://www.mdpi.com/1999-4915/14/5/1035

https://www.ncbi.nlm.nih.gov/pmc/articles/PMC8739610/

https://www.ncbi.nlm.nih.gov/pmc/articles/PMC8806134/

https://dergipark.org.tr/tr/download/article-file/1926629

Author Response

  • Response to Comments from Reviewer 3

Comment 1:

  1. Does the meta-analysis include the patient infection stage (short, long, acute...)? Why the long Covid-19 patents were excluded?

Response:            

Thank you for the comment. In this revised manuscript, we described the definition of long COVID and added why it is excluded from this review.

“2.1.2. Types of participants. Only patients infected with SARS-CoV-2 were included. This review was intended to focus on the direct impact of SARS-CoV-2 infection in COVID-19 patients, and thus long COVID (i.e., the continuation or development of new symptoms 3 months after the initial infection with SARS-CoV-2) patients were excluded. This is because long COVID does not necessarily require the current presence of SARS-CoV-2, and the condition is not dependent from the severity of acute SARS-CoV-2 infection [21]. Therefore, the association between long COVID and HRV would be worthy of being investigated as a separate topic. No restrictions were imposed on clinical condition, language, sex/gender, age, or race/ethnicity.”

(Please see in page 2, red words)

Comment 2:

  1. Which methods were used among studies for data extraction and filtration in HRV parameters (especially in FitBit and other non-medical devices)? Are the data comparable in terms of quality?

Response:            

Thank you for the comment. Heterogeneity of studies, including how HRV parameters were measured, is one of the reasons for not conducting a quantitative analysis in this review. In this revised manuscript, we analyzed the heterogeneity of the included studies with respect to the HRV measurement method and described the related limitations.

“The HRV measurement method also showed differences according to the included studies. In particular, nine studies [49,51,54-57,60,61,65] (9/17, 52.94%) used ECG to measure HRV parameters, and the measurement time also varied from 10 seconds to 24 hours. Only three studies [49,60,61] (3/9, 30%) described obtaining HRV parameters with 24 h of holter recording. Meanwhile, four studies [52,53,62,64] (4/17, 23.53%) reported using commercially available wearable devices, and one study [58] reported using a wearable wireless sensor, but it was unclear whether they were commercially available (Table 1).”

(Please see in pages 13-14, red words)

First, the heterogeneity of the studies included in this review is a major limitation of this review. Our review qualitatively analyzed the heterogeneity of the included studies, and found considerable heterogeneity in the study design, clinical characteristics of participants, and HRV measurement method. Among the included studies, the limitation that information on the cardiovascular conditions of the participants could be confirmed in only nine studies [49-51,54,55,58,60,61,63] (9/17, 52.94%) may also contribute to the heterogeneity. As, only 53% of included studies [49,51,54-57,60,61,65] used ECG to measure HRV parameters, and it cannot be considered to have the equal reliability as the HRV parameters obtained for commercially available wearable devices such as Fitbits. In addition, it is still possible that the heterogeneity was due to the age or sex of the participants, the difference in their underlying diseases, the difference in the duration of COVID-19, the difference in the history of COVID-19 vaccination, and the change in HRV due to factors other than SARS-CoV-2 infection. As the meta-analysis also reaffirmed the substantial heterogeneity included (I-square values, 60.0% to 96.5%), the meta-analysis was not interpreted clinically.”

(Please see in page 16, red words)

Comment 3:

  1. The Results section is not systematized. The reader encounters many opposite results, comparing study by study. Results, Discussion, and Conclusion must be systematized in order to address some of the main criteria of SARS-CoV-2 infection impact on HRV among selected studies.

Response:            

Thank you for the comment. We apologize for the inconvenience caused by the unsystematized manuscript. We have tried to systematize the structure of the manuscript as much as possible in this revised manuscript. Nevertheless, if this revised manuscript still seems unsystematic to the reviewer, we are willing to make further revisions.

(Please see in page 3, red words)

Comment 4:

The main results and differences among time and frequency HRV domains should be highlighted.

What is the common denominator of all the studies?

Heterogeneity did not bring clarity to the presentation of the results.

Response:            

Thank you for the comment.

1) In this revised manuscript, we differentiated into vagally mediated HRV (vmHRV) parameters and other HRV parameters to make it easier to interpret HRV parameters clinically.

“Considering the heterogeneity of the population, times from the COVID-19 outbreak, and potential comorbid diseases, quantitative analysis was not planned in the protocol of this systematic review. A quantitative analysis can only be performed if sufficient homogeneity between the studies and outcomes used is ensured. However, the clinical heterogeneity between the included studies was considerable. Therefore, the impact of SARS-CoV-2 infection on HRV was analyzed qualitatively. Among the HRV parameters, clinically relevant vagally mediated HRV (vmHRV) parameters were of interest, which include RMSS, HF power, and HFnorm.”

(Please see in page 4, red words)

“3.4.2. Impact on vmHRV parameters

(1) SARS-CoV-2 infection (vs. negative control): …”

(Please see in page 11, red words)

Figure 2. Impact of COVID-19 on vmHRV parameters.”

(Please see in page 12, red words)

“4.1. Findings of this review

This review was conducted to systematically investigate the effect of SARS-CoV-2 infection on HRV …

(3) Impact on vmHRV parameters: …”

(Please see in page 14, red words)

2) In the included studies, the common denominator was largely divided into two. That is, a comparison of SARS-CoV-2 infected and non-infected patients, and patients at different stages of COVID-19.

“We hypothesized that patients infected with SARS-CoV-2 show significant differences in HRV-related outcomes compared to non-infected individuals. There was also a hypothesis in this review that HRV parameters would be different depending on the severity and prognosis of COVID-19.”

(Please see in page 2, red words)

“2.1.4. Types of controls. Non-infected individuals, healthy controls, or infected patients of different severity of COVID-19 were included in the control group. For longitudinal studies, the time of infection, including pre-infection, was included. As a relationship between SARS-CoV-2 infection and HRV-related parameters has not yet been well established, this review focused on the impact of SARS-CoV-2 infection on HRV parameters. And comparative studies of COVID-19 patients and individuals with other infectious diseases were outside the scope of this review. Specifically, comparisons between COVID-19 patients and non-infected individuals or healthy controls investigated the association between SARS-CoV-2 infection and HRV parameters. On the other hand, comparisons among COVID-19 of different severities investigated the clinical usefulness of HRV in the context of assessment of patients with COVID-19.”

(Please see in page 3, red words)

“3.4.2. Impact on vmHRV parameters

(1) SARS-CoV-2 infection (vs. negative control): Mixed results were observed …

(2) Different severity of COVID-19: In relation to the different severity or prognosis of COVID-19, there were some studies that have found significant …”

(Please see in page 11, red words)

“3.4.3. Impact on other HRV parameters

(1) SARS-CoV-2 infection (vs. negative control): Regarding SDNN (ms), …

(2) Different severity of COVID-19: Significant differences in some …”

(Please see in pages 12-13, red words)

3) The heterogeneity of the included studies in this reviewed manuscript was further explored qualitatively.

“2.7. Assessment of heterogeneity

The heterogeneity of included studies was investigated both qualitatively and quantitatively. Qualitatively, the potential causes of heterogeneity of the included studies according to study design, clinical characteristics of participants, and HRV measurement method were analyzed. Moreover, the authors performed a meta-analysis for the purpose of confirming the justification of performing only qualitative analysis.”

(Please see in page 4, red words)

“3.6. Assessment the heterogeneity

3.6.1. Qualitative analysis. The heterogeneity of the included studies was analyzed qualitatively in terms of the study design, clinical characteristics of participants, and HRV measurement method. The study designs of the included studies varied to a total of four (i.e., case-control study, retrospective analysis, cross-sectional study, and prospective cohort study), and only four included studies [50,53,62,64] (4/17, 23.53%) were prospective cohort studies with relatively strict study designs. In addition, only five studies [55,58,59,64,65] (5/17, 29.41%) were identified that statistically adjusted key potential confounding variables in their analysis.

Regarding the clinical characteristics of participants, most included studies [49,51-55,58,61,62,64,65] (10/17, 58.82%) recruited and classified participants according to SARS-CoV-2 positivity. However, they did not categorize according to COVID-19 severity, so it is possible that COVID-19 patients of varying severity were included. Meanwhile, some studies have classified COVID-19 patients according to clinical severity [59,60], recruited only critically ill patients [50,56], or classified participants with specific clinical sign including silent hypoxia [57] and death [50,63]. In addition, one study was interested in occurrence of COVID-19 in cAF patients only [54]. One study [65] investigated the impact of the Omicron variant of COVID-19, but other studies did not limit the variant.

The HRV measurement method also showed differences according to the included studies. In particular, nine studies [49,51,54-57,60,61,65] (9/17, 52.94%) used ECG to measure HRV parameters, and the measurement time also varied from 10 seconds to 24 hours. Only three studies [49,60,61] (3/9, 30%) described obtaining HRV parameters with 24 h of holter recording. Meanwhile, four studies [52,53,62,64] (4/17, 23.53%) reported using commercially available wearable devices, and one study [58] reported using a wearable wireless sensor, but it was unclear whether they were commercially available (Table 1).

3.6.2. Quantitative analysis. Meta-analysis in comparison of COVID-19 positive patients and negative controls was possible in five HRV parameters from five studies [49,55,59,61,65], including two vmHRV (i.e., RMSSD and HF power), and three other parameters (i.e., SDNN, LF power, and LF/HF ratio). Meanwhile, meta-analysis in the comparison of symptomatic COVID-19 patients and asymptomatic controls was possible in two parameters from two studies [49,55], including SDNN and LF/HF ratio. Substantial visual and statistical heterogeneity (I-square values, 60.0% to 96.5%) was found in the meta-analysis results. Among the results of the meta-analysis, only the vmHRV parameter showed a statistically significant difference, that is, RMSSD (ms) was significantly greater (SMD, 1.15; 95% CIs, 0.60 to 1.71; p = 0.030) and HF power (ms2) was significantly lower (SMD, -1.49; 95% CIs, -2.64 to -0.34; p = 0.000) in COVID-19 patients compared to negative controls (Supplementary file 4).”

(Please see in pages 13-14, red words)

Comment 4:

  1. The methodological research does not include studies like the following:

https://www.mdpi.com/1999-4915/14/5/1035

https://www.ncbi.nlm.nih.gov/pmc/articles/PMC8739610/

https://www.ncbi.nlm.nih.gov/pmc/articles/PMC8806134/

https://dergipark.org.tr/tr/download/article-file/1926629

Response:            

Thank you for the comment. The reasons for each study not included in this review are as follows:

It differs from the inclusion criteria of this study described (i.e., for Long COVID patients or those who have recovered from COVID-19).

https://www.mdpi.com/1999-4915/14/5/1035

https://www.ncbi.nlm.nih.gov/pmc/articles/PMC8739610/

https://www.ncbi.nlm.nih.gov/pmc/articles/PMC8806134/

The following study was not retrieved from our searches. Also, we did not find this study in the reference list of related reviews either. We have added this study to this review in this revised manuscript.

https://dergipark.org.tr/tr/download/article-file/1926629

“Therefore, 17 observational studies (total participants, N = 3628) [49-65], including a study from other sources [61], were included in this review (Figure 1).”

(Please see in page 4, red words)

Round 2

Reviewer 1 Report

Dear authors,

your manuscript underwent extensive revision, and as far as I can tell it improved a lot. I have no other iquiries according your work. Congratulations.

Reviewer 3 Report

The authors have addressed all questions with appropriate changes.